# Subjective and objective measures of visual awareness converge

**Markus Kiefer ⓘ \*, Verena Frühauf, Thomas Kammer ⓘ**

Department of Psychiatry, Ulm University, Ulm, Germany

\* Markus.Kiefer@uni-ulm.de

## Abstract

Within consciousness research, the most appropriate assessment of visual awareness is matter of a controversial debate: Subjective measures rely on introspections of the observer related to perceptual experiences, whereas objective measures are based on performance of the observer to accurately detect or discriminate the stimulus in question across a series of trials. In the present study, we compared subjective and objective awareness measurements across different stimulus feature and contrast levels using a temporal two-alternative forced choice task. This task has the advantage to provide an objective psychophysical performance measurement, while minimizing biases from unconscious processing. Thresholds based on subjective ratings with the Perceptual Awareness Scale (PAS) and on performance accuracy were determined for detection (stimulus presence) and discrimination (letter case) tasks at high and low stimulus contrast. We found a comparable pattern of thresholds across tasks and contrasts for objective and subjective measurements of awareness. These findings suggest that objective performance measures based on accuracy and subjective ratings of the visual experience can provide similar information on the feature-content of a percept. The observed similarity of thresholds validates psychophysical and subjective approaches to awareness as providing converging and thus most likely veridical measures of awareness.

## 1. Introduction

Phenomenal consciousness [1], which refers to the experiential qualities of sensations, imposes serious challenges for its scientific investigation due to the first-person perspective and thus private character of subjective experiences [2]. Despite these intrinsic difficulties, a reliable and valid assessment of this phenomenon is crucial for determining the underlying neuro-cognitive mechanisms [3,4].

As one aspect of phenomenal consciousness, the emergence of visual awareness has been subject of several studies in the past [5–9]. In a prototypical experiment, observers are presented with briefly presented visual target stimuli, which are followed by a pattern mask consisting of for instance randomly arranged visual elements [e.g., 8,10]. Systematic variation of target-mask stimulus onset asynchrony (target-mask SOA) serves to induce different states of stimulus awareness in the observer, because the mask interferes with the consolidation process within the visual system at different time points. When in such experiments stimulus

**Funding:** The authors received no specific funding for this work.

**Competing interests:** The authors have declared that no competing interests exist.

awareness of the observers is assessed, at short target-mask SOAs observers have typically little or no awareness of the stimuli, whereas they are more likely aware of the stimuli at longer SOAs.

Within consciousness research, the most appropriate assessment of visual awareness, subjective vs. objective measures of awareness, is matter of a controversial debate [5,8,10–15]. Subjective measures rely on introspections of the observer related to perceptual experiences induced by the visual stimuli [7,10,15,16] such as categorical classifications of the percept ("seen" or "unseen", [6]), graded ratings of the clarity of the percept [15,17], post-decisional wagering [18] or confidence [8,14,19] as well as phenomenological reports of the sensations [7,20]. For the purpose of the present study, we focus on the perceptual awareness scale (PAS), which requires observers to judge their perceptual experience on a four-point rating scale ranging from complete unawareness to full awareness [15,16]: 1 = "I do not see the stimulus at all"; 2 = "I saw a glimpse of something, but don't know what it was"; 3 = "I saw something, and I think I can determine what it was", 4 = "I saw the stimulus clearly".

Objective measures of visual awareness are based on a psychophysical approach, which assesses performance of the observer to accurately detect or discriminate the visual stimulus in question across a series of trials [21–23]. Besides directly analyzing performance accuracy, indices putatively measuring visual awareness are derived from the accuracy distribution across trials [22,24]. For instance, sensitivity indices such as d' based on signal detection theory can be calculated from hit and false alarm rate to separate sensitivity from response bias [25]. Alternatively, psychophysical functions [26] can be fitted to the entire accuracy distribution, and thresholds as well as slopes of the psychophysical function are determined to characterize awareness of the observers [5,10,18].

Some researchers argue that subjective measures of awareness such as ratings of the clarity of the percept should be preferred over objective psychophysical performance measures, because the former directly capture experiential qualities of sensations in line with the subjective nature of phenomenal consciousness [e.g., 11,16,17,27]. Phenomenal consciousness is defined as the subjective perceptual experience of a stimulus and involves a first person perspective [1]. Objective performance measures reflect access consciousness, i.e. accurate responding in congruency with task instructions or action goals [1]. Access consciousness is therefore defined as information processing leading to supra-threshold accurate task performance (e.g., response to stimulus presence or letter case) and involves a third person perspective.

Subjective measures based on introspection are furthermore considered as advantageous, because objective measurements of detection or discrimination performance can be influenced by unconscious processing and therefore may not exclusively index visual awareness [10,11,18]. Objective measures might also be less exhaustive than subjective measures, because not all perceptual experiences induced by a stimulus might contribute to performance [7,28]. Furthermore, it has been argued that objective measures also do not capture trial-wise fluctuation of awareness [29]. Finally, as objective awareness measures based on signal detection theory typically average detection or discrimination performance over a series of trials, they do not consider the entire performance distribution. This latter criticism, however, does not apply to threshold and slope measurements determined from psychophysical functions fitted to the accuracy distribution [5].

However, the validity of subjective ratings of introspective experience in single trials has also been critically discussed: Subjective ratings in general might be affected by response biases and cannot be necessarily taken as direct reflections of subjective visual experience [13,22,24,30]. Observers might also use the PAS scale in a task-dependent fashion suggesting that the rating categories cannot be uniformly interpreted [28,31]. Furthermore, the PAS

scale might not specifically reflect clarity of the visual experience, but also non-visual content such as intuitions, or confidence [14,19,32,33]. Finally, visual sensations induced by a barely visible masked stimulus presented at threshold can include experiences of movements, rotations or stimulus expansions [7,20], which may not be precisely captured with a one-dimensional scale [7].

These difficulties associated with the interpretation of objective vs. subjective measures of awareness are increased by observations that both measures are sometimes not linearly correlated suggesting that they capture different cognitive states [e.g, 31,32]. In addition, even if they were correlated in some studies, subjective measures of awareness were temporally delayed compared to performance measures [10,28]: For instance, Sandberg and colleges [10] reanalyzed the data of an earlier study [8], in which four simple geometrical objects (pictures of a circle, square, triangle or diamond) were briefly presented at 12 different stimulus durations (possible durations varied between 16 and 192 ms in steps of 16 ms) and thereafter masked with a stimulus consisting of all four shapes. Participants were first asked to indicate the displayed shape as fast and accurately as possible, prioritizing accuracy over speed. Thereafter, participants reported their subjective awareness using either PAS, a confidence rating scale or a post-decision wagering scale. In their reanalysis, Sandberg and colleagues fitted psychophysical functions to both objective performance accuracy and subjective awareness ratings (PAS, confidence scale, wagering scale) as a function of stimulus durations (which also results in corresponding changes of mask stimulus SOAs). They consistently found a horizontal shift between accuracy curves and curves based on subjective awareness ratings, with the largest shifts for PAS curves [10]. This suggests that subjective awareness thresholds including those based upon PAS ratings were higher than objective thresholds based upon performance accuracy. This temporal lag between subjective awareness and objective performance thresholds have been taken as indication that performance within this temporal lag is mainly driven by unconscious processing [see also 18]. If this interpretation is correct, a psychophysical approach to consciousness based on objective performance measurements would be largely meaningless due to their contamination by unconscious processing [11].

However, it seems premature to abandon a psychophysical approach to consciousness based on the data reported above for several reasons. Objective performance could be measured with an appropriate task, which minimizes responding based on unconscious processing. Furthermore, precision of measurements of subjective and objective awareness thresholds could be improved by an individual adaptation of the visual stimulation (target-mask SOA), which probes the entire range of awareness states from complete unawareness to full awareness at an objective and subjective level in each individual observer. In previous studies comparing objective and subjective awareness measures, observers received a constant variation of target-mask SOAs, which might be not sufficient to elicit the entire range of states of awareness in each observer [e.g., 10]. Finally, instructions in the study by Sandberg and colleagues [10] stressed response speed in addition to accuracy in the objective task, while there was no speed instruction for the PAS ratings as it is typical for ratings in general. It is possible that the speed instruction in this earlier study additionally induced responding based upon unconscious response priming in the objective task [34].

In order to test the relation between objective and subjective measurements of visual awareness in a condition without the limitations mentioned above, in the present study (study protocol preregistered at https://osf.io/w4rqg), we determined thresholds for visual word perception at an objective performance and a subjective judgment level using a temporal two-alternative forced choice task (temporal 2-AFC). In the task, participants were presented with two stimulus-mask sequences separated by 900 ms (see Fig 1). The critical task-relevant stimulus feature (e.g., presence vs. absence of a stimulus or uppercase vs. lowercase letters) randomly

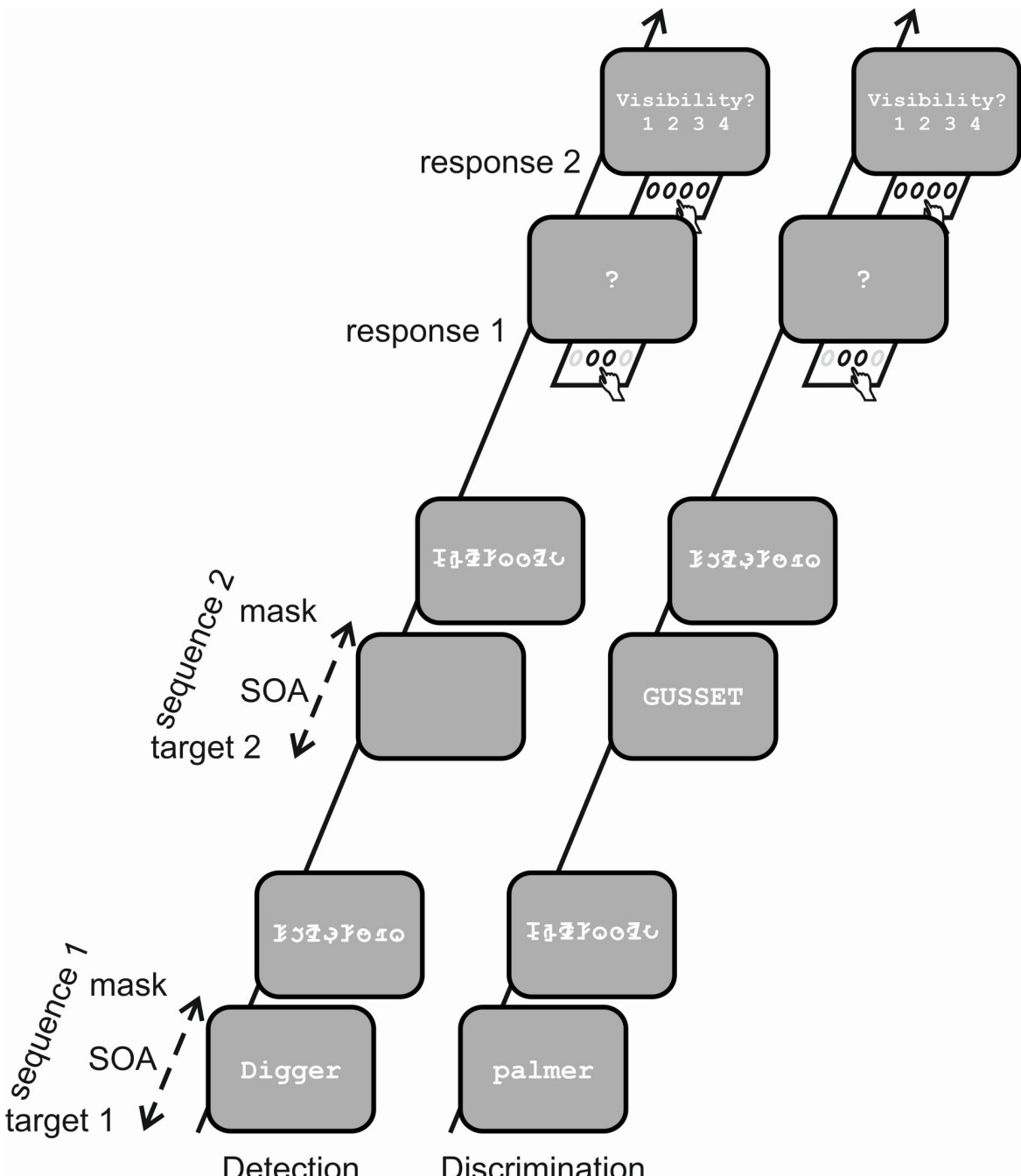

**Fig 1. Time course of the tasks.** Each of the two tasks consists of two target-mask sequences with a target (one frame) followed by a mask formed by a string of false font letters (200 ms). Stimulus onset asynchrony (SOA) between target and mask was similar in both sequences, adaptively varied in a range between 6.7 ms and 340 ms in order to measure objective and subjective perception thresholds. The interval between the two sequences was 900 ms. After the second sequence two questions were asked subsequently: (1) In what interval was the target flashed? (indicated by a question mark); (2) What was the visibility like? (PAS-scale, 1–4). In the detection task in one of the two sequences a word was flashed, the other sequence consisted of a mask only. The discrimination task consisted of two target words, one written in capital letters, the other in small letters.

appears either in the first or the second interval. After the second sequence, participants are prompted to indicate in which interval the designated stimulus feature was presented.

In order to compare subjective and objective awareness measurements across different stimulus feature and task difficulty levels, thresholds were determined for detection (stimulus presence) and discrimination (letter case) tasks at high and low stimulus contrast. The temporal 2-AFC task has been previously employed to demonstrate the context- and feature-dependency of objective awareness thresholds [5,35]. This task has the advantage to provide an objective psychophysical performance measurement, while minimizing biases from unconscious processing. The required comparison of the percepts in the two intervals cannot be performed based on unconscious visual processes such as unconscious response priming [36]. A further advantage of the temporal 2-AFC task is that it always involves a true two-alternative choice reaction ("Is the critical feature in the first or in the second interval?") irrespective of the complexity of the probed feature (stimulus presence vs. letter case). In a classical non-temporal decision task with only one stimulus interval, the detection of stimulus presence cannot be considered to be a true 2-AFC task, because response alternatives "stimulus present" vs. "stimulus absent" are not equally weighted [21 p. 27]. Furthermore, unlike in the study by Sandberg and colleagues [10] our instructions only stressed response accuracy, but not response speed in the objective task, in order to further minimize response base upon unconscious response priming.

As a novel aspect in the present study, subsequent to the task response measuring objective performance, observers judged their perceptual experience of the masked stimuli using PAS ratings. Before the main experiment, observers were intensively trained to relate their phenomenological impressions to the categories of the PAS so that they use the scale in the intended manner to report their sensations.

Visibility of the word stimuli was manipulated and adapted to each observer by gradually varying the target stimulus mask onset asynchrony (target-mask SOA) according to a staircase algorithm. In order to obtain an appropriate sampling for both accuracy and PAS ratings across performance and ratings levels, separate staircases related to accuracy and PAS ratings, respectively, were implemented. Psychophysical functions were then fitted to the distribution of accuracy data and PAS ratings in the detection (stimulus presence) and in the discrimination (letter case) tasks as a function of the stimulus mask SOA. Subjective (PAS ratings) and objective (accuracy data) conscious detection or discrimination SOA thresholds were than determined (for details see the methods section). Note that subjective detection and discrimination thresholds map onto different levels of the PAS, if the scale is used in the instructed manner. Stimulus detection refers to a visual impression described by PAS level 2 ("I saw a glimpse of something, but don't know what it was"); whereas stimulus discrimination refers to a visual impression described by PAS level 3 ("I saw something, and I think I can determine what it was"). In analogy to the objective threshold definition of 0.75, i.e. the transition between pure guessing (0.5) and correct response (1.0) in the 2-AFC task, we therefore defined the subjective thresholds from the fitted psychophysical functions in the stimulus detection task at PAS level 1.5 (transition between "nothing" and "glimpse") and in the stimulus discrimination task at PAS level 2.5 (transition between "glimpse" and "I saw something"). This aspect of data analysis is not mentioned in the preregistration, but was determined before data collection had started. The objective thresholds were taken from the fitted functions at an accuracy of 75% in all tasks (inflection point of the curve), as it is common in psychophysical research [cf. 21].

A higher threshold indicates that the stimulus has to be presented longer in isolation before mask onset so that a given task-relevant feature can be consciously identified at a subjective or objective level. In addition to thresholds, slopes of the fitted psychophysical functions in the

different conditions were estimated using the width parameter (interval between 5% and 95% quantile). A large width (i.e., smaller steepness of the function) has been taken to index a more gradual transition from unconscious to conscious at the feature level, whereas a small width (i.e., larger steepness of the function) indexes a more abrupt or dichotomous transition [10,18]. Alternatively, smaller steepness of the psychometric function might also result from an abrupt transition, which however temporally varies on a trial to trial basis. Averaging across many such trials might lead to a flattening of the psychometric function. Note, however, that a potential substantial presence of intermediate PAS ratings (PAS levels 2 and 3), which indicate states of partial awareness, renders the interpretation in terms of variable dichotomous transitions unlikely.

Based on proposals assuming a gradual emergence of consciousness [11,16,37] as well as the partial awareness hypothesis [9,27,38] and previous findings [5], we assume that awareness of stimulus presence require a shorter time of stimulus consolidation within the visual system than awareness of letter form [for comparable time course difference between detection and identification, see also 35,39,40]. We therefore expect lower thresholds for stimulus presence/ absence decisions than for letter case decisions in the capital letter discrimination task. Similarly, high contrast stimuli, due to their higher physical energy, should be faster consolidated than low contrast stimuli [41] and therefore be visible at lower thresholds. Furthermore, as visual consolidation is assumed to be generally fast at high stimulus contrast, threshold differences between stimulus presence decisions and letter form discriminations should be reduced at high than at low contrast [42]. The differential speed of visual consolidation at the feature and contrast levels should also be reflected by the slopes of the psychophysical function, which index the transition from unawareness to awareness within each feature type [18]. Based on previous studies [5,35], we expect steeper slopes, i.e. more discontinuous emergence awareness, for stimulus presence decisions than for letter form discriminations as well as for high contrast than for low contrast stimuli. This assumption deviates from earlier findings in the context of the levels of processing approach [27,28], which proposes shallower slopes for lower-level tasks and steeper slopes for higher-level tasks. As already outlined earlier [5], we assume that slopes of psychophysical functions and thus the transition from unawareness to awareness for a given feature are highly flexible and depend among other factors on the dimensional continuity of the features probed in the task [see also, 9]: In the present study, the dimension of stimulus absence vs. presence is highly discontinuous, whereas the dimension of letter form based on letter case is more continuous. We therefore propose that awareness of a feature on a continuous dimension emerges temporally more gradually compared with features on a discontinuous dimension, which give rise to sharp transitions between unaware and aware states.

With regard to the relation between objective performance and subjective PAS ratings measures, depending on the precise theoretical stance, three alternative hypothetical scenarios are possible: Firstly, thresholds derived from objective and subjective measures of awareness could be unrelated across tasks and contrasts, because these measures might capture qualitatively different forms of information or representations [31,32] linked to access vs. phenomenal consciousness [1,43]. Objective and subjective measures may also partially depend on different types processes [for a discussion, also see 44]: For instance, only subjective, but not objective measures might depend on second-order meta-cognitive processes, which evaluate the representation [33]. Secondly, objective and subjective thresholds could exhibit a comparable pattern as a function of task and contrast, but subjective thresholds would be temporally delayed by a constant lag. This lag may arise because above-threshold objective performance might be partially based on fast unconscious processing, while above-threshold subjective ratings may require longer lasting visual consolidation giving rise to a specific phenomenal experience

[10,18,28]. Thirdly, PAS ratings and accuracy of task performance could comparably reflect the phenomenal content of visual awareness. As a consequence, thresholds should be similar for subjective and objective measures, exhibiting a zero-lag.

## 2. Methods

### 2.1. Subjects

Twenty-three (mean age 21.5, 13 female) native German-speaking volunteers without any history of neurological or psychiatric disorders were recruited for the study. They participated after giving written informed consent and were compensated for participation either by money or by course credits. The study has been approved by the institutional review board of Ulm University and was performed in accordance with the regulations of the Declaration of Helsinki. Visual acuity was tested using FrACT [Version 3.9.9, central Landoldt-C, 4-AFC, 18 trials, observer distance 2.5 m, 45]. All participants had a binocular visus of 1.15 $VA_{dec}$ at minimum. Three participants were excluded from data analysis; in one subjective thresholds could not be estimated due to inconsistent responses. In two more participants, accuracy in the visibility task was above chance performance (see below). This exclusion criterion was not mentioned in the preregistration, but is consistent with our previous work [5]. Mean age of the remaining 20 participants (11 females) entering data analyses was 21.7 (range 19–25 years). Hence, data of one more participant entered data analyses as determined in the power analysis of the preregistration, because dropout was less than expected. This power analysis using GPower [46] indicated that a sample size of n = 19 is needed to achieve a power of β = .90 to detect the TASK x CONTRAST interaction (α = 05, two-tailed). Power analysis was based on an effect size of $\eta_p^2$ = 0.39, which was derived from an unpublished pilot study. Conversion of this effect size measure according to the formula described in the tutorial by Langenberg and colleagues [47] approximately yields a d (or f) = 0.80.

### 2.2. Stimuli and apparatus

Experimental setup was the same as described in an earlier study [5]. In short, stimuli were presented on a CRT screen (21', iiyama, Hoofddorp, The Netherlands) at a frame rate of 150 Hz using Psychopy [v1.78.01, cf.48]. Target words (six letters) with a luminance of 16 cd/m$^2$ (low contrast) or 25 cd/m$^2$ (high contrast) were flashed for one frame with a height of 0.38˚ in white (25 cd/m$^2$) on a grey background (5 cd/m$^2$). The mask with a duration of 30 frames (200 ms) followed the target with an SOA varying from 1 to 50 frames, i.e. 6.7–340 ms. It consisted of an 8 symbol random string from a false font and had a luminance of 50 cd/m$^2$.

Subjects sat in front of a CRT screen with a distance of 1.5 m in a room with dimmed ambient light. During the experiment, they responded with left and right index fingers, respectively, on a keyboard with 4 keys. Key assignment to the response categories was counterbalanced across participants.

Five lists of words with 6 characters were used [5]. Two lists consisted of 76 words each, applied for stimulus presence/absence decision (see below). In case of letter form decision two more lists with 76 pairs of words were used, one written in lowercase letters, the other in uppercase letters. A fifth list with 50 single words was applied for the final visibility test.

### 2.3. Procedure

Participants performed four different temporal two-alternative forced choice tasks (2 intervals) with 76 trials each, in which objective and subjective detection or discrimination thresholds were measured, by adaptively varying the target-mask SOA. A trial consisted of the

presentation of two target-mask sequences, finished by a temporal two-alternative forced choice decision (objective measurement) and a subjective PAS rating. The sequence started with a hash (#) for 500 ms followed by a blank screen for 500 ms. Then the target word was flashed for one frame followed by the mask for 200 ms. Target and mask were separated by a blank period with a variable SOA of 6.7–340 ms. After a pause (blank screen for 900 ms following the presentation of the mask), the second sequence with identical timing parameters started with a hash. Immediately after the end of the second sequence, a question mark indicated the subject to respond for the sequence with the stimulus feature asked for (1 or 2). Then, to prompt the PAS rating the message "Visibility? 1 2 3 4" was displayed. (see Fig 1, for sake of clarity the two hashes are not depicted). Three randomly interleaved simple staircases (2 correct answers: SOA down; 1 incorrect answer: SOA up) were used to adaptively vary SOA over the 76 trials. Two of them were controlled by objective responses. They started at 20 ms and 40 ms, with a step size of 13 ms each. The third staircase covered longer SOAs and was controlled by subjective responses. The "correct" criterion was a response of 4, and response below 4 was considered as "wrong". This staircase started at 73 ms with a step size of 26.7 ms (Fig 2). Together the three staircases covered the full range of the psychometric function for both objective and subjective responses, despite restricted stimulus material (76 presentations per task). In each trial subjects had to respond to the two different questions, independent of the actual staircase.

The perceptual awareness scale (PAS) consisted of level 1–4 (see introduction). Prior to the first task subjects were systematically trained in applying PAS with the stimulus presence task at high contrast. First, a long SOA was chosen in order to demonstrate full visibility (level 4). By manually shortening SOA PAS conditions level 3, 2, and 1 were evoked, and by asking for the full word it was clarified that the ability to identify the word should be rated with 4, whether awareness of only some letters without the full word should be rated with 3. Similarly, each glimpse was trained to be rated with 2, so that level 1 was only used in case of full invisibility. In a second step PAS was trained with the capital letter decision task in the same manner. Training was terminated when subjects were able to consistently apply PAS. As a consequence, a correct response in the decision task is already possible with a PAS rating of 3 (awareness of letter form), whereas a correct response in the detection task requires a PAS rating of 2 (glimpse) or more. At least 16 trials were used for training. PAS training was not combined with any training to objective responses.

The order of the four tasks was systematically permutated over subjects. It started either with the detection task ("In which interval a word was flashed?") or with the letter discrimination task ("In which interval the word was written in capital letters?") using one contrast level. Subsequently the same task with the second contrast level was applied, followed by the two tasks of the second condition. At the end of the experiment, a visibility task was applied. Similar to the absent/present task, in only one sequence of the two-interval forced choice task a target word was presented at high contrast. In difference to the former tasks the target-mask SOA was fixed to 6.7 ms, and no discrimination threshold but a detection ratio was measured. This task served to assess whether stimuli were entirely unconscious at minimum stimulus mask SOA. According to binomial distribution, a correct stimulus detection rate of 32/50 trials [64%] or more was defined as performance above chance level (p = 0.03), thus, a complete masking of the stimulus was not achieved in these subjects.

### 2.4. Data analysis

In each individual and task, responses from the three staircases were collapsed, and a psychometric function (Fig 2) was fitted to the accuracy distribution as a function of target-mask

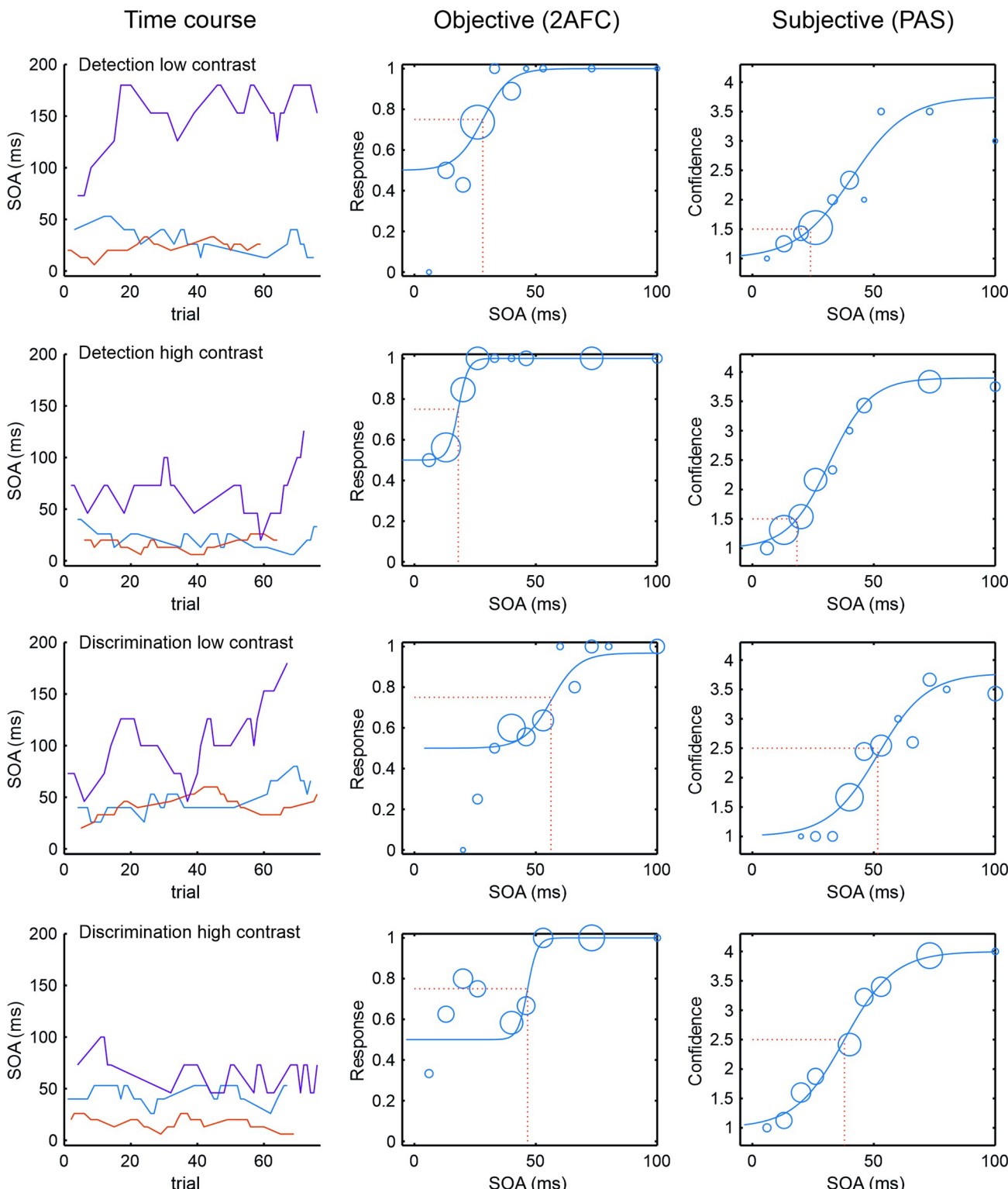

**Fig 2. Analysis of raw data.** Results from one subject are shown. In the left column the time course of the experiment is depicted. Three staircases randomly interleaved started at different SOAs (red 20 ms, blue 40 ms, and purple 73 ms). Trial number was fixed at n = 76. Sigmoidal fits over the answers merged from the 3 staircases (circles) are shown in the middle column (objective responses, temporal 2-AFC) and right column (PAS ratings). Number of observations per SOA is depicted by the radius of the circles. Each of the four rows displays a task condition as indicated in the left column. The range of objective responses is between 0.5 (guessing rate) and 1 (no errors), the PAS ratings vary between 1 and 4. Please notice that subjective threshold definition differs between detection task (1.5) and discrimination task (2.5). For sake of clarity fits are only depicted between 0 and 100 ms SOA.

SOA as well as to the PAS rating distribution using psignifit [v 4.0, cf.49]. The logistic function

$$F(x) = \gamma + (1 - \gamma - \lambda) \frac{1}{1 + e^{-2 \log\left(\frac{1}{0.05} - 1\right) \frac{x - m}{w}}}$$

was applied, with $m$ as threshold (point of inflection of the logistic function), $w$ as width (x-range from 5% to 95%), $\gamma$ as probability rate, and $\lambda$ as lapsus rate (free within a range from 0 to 0.3). In case of objective thresholds (2-AFC) $\gamma$ was fixed to 0.5 (guessing rate) and $m$ was taken as threshold (SOA with 75% correct responses). Fixation of $\gamma$ was mandatory since due to the restriction of stimulus material we did not collect a sufficient number of data points at low SOAs in order to empirically confirm a guessing rate for the 2-AFC task. However, demonstration of invisibility with the last task justifies the assumption of $\gamma$ at 0.5. In case of subjective thresholds (PAS) data were transposed to a range from 0 to 1, and $\gamma$ was fixed to zero. Threshold for discrimination (PAS of 2.5) was read out as $m$ (0.5, point of inflection), and threshold for detection (absent/present task) was read out on the estimated function at PAS of 1.5 (0.167 on the transposed function), the transition from "nothing at all" (1) and "glimpse" (2). Thresholds and slopes were analyzed using a repeated-measure analysis of variance (ANOVA, Statistica V13.3, StatSoft, Hamburg, Germany). Post-hoc analyses were performed using Newman-Keuls tests. In addition, a Bayesian t-test was calculated for thresholds [JASP 0.16.2. JASP Team 2022, 50]. The latter aspect of the analyses was not mentioned in the preregistration.

## 3. Results

### 3.1 Histogram of raw data

For each condition and contrast level raw data from all subjects were collapsed to a scatter-icon histogram, where responses were summed up by SOA and subjective response (PAS-level). For each element the amount of correct and wrong responses in the objective task is depicted as pie-diagram (Fig 3). Comparing the four conditions it is obvious that subjective thresholds are lower with high contrast, and they are lower for detection tasks with respect to discrimination tasks. Furthermore, differences in width can be seen in the cumulative data. Width is larger in discrimination tasks with respect to detection tasks, and width is larger at low contrast compared to high contrast. Regarding the ratio of objective responses in the detection tasks only lowest SOAs at PAS level 1 ("nothing seen") show about 50% of correct responses, the guessing probability. With higher SOAs the ratio of correct responses reaches about 70%. At PAS level 2 ("glimpse") with higher SOAs correct responses reaches 90% -100%, whereas at PAS level 3 and 4 wrong responses were extremely rare. This is completely different with the discrimination tasks. Here at PAS level 1 with most of the SOAs guessing probability is observed. At PAS level 2 a transition from guessing rates to about 75% correct is seen with higher SOAs. Still at PAS level 3 ("I saw something") only up to 80% correct responses are reached, whereas at PAS level 4 ("clearly seen") correct responses range between 80% and 100%.

### 3.2 Analysis of threshold

Threshold data were subjected to a repeated-measures ANOVA with the factors TASK (absent/present and capital), CONTRAST (low and high) and MODE (objective and subjective). This analysis (Fig 4, S1 Table in S1 File) yielded significant main effects of the factors TASK [$F_{(1,19)} = 194$, $p < .001$, $\eta_p^2 = .911$] and CONTRAST [$F_{(1,19)} = 274$, $p < .001$, $\eta_p^2 = .935$], whereas the factor MODE was not statistically reliable [$F_{(1,19)} = 0.78$, $p = .388$, $\eta_p^2 = .039$]. Thresholds for the detection task (words absent or present) were lower compared to the

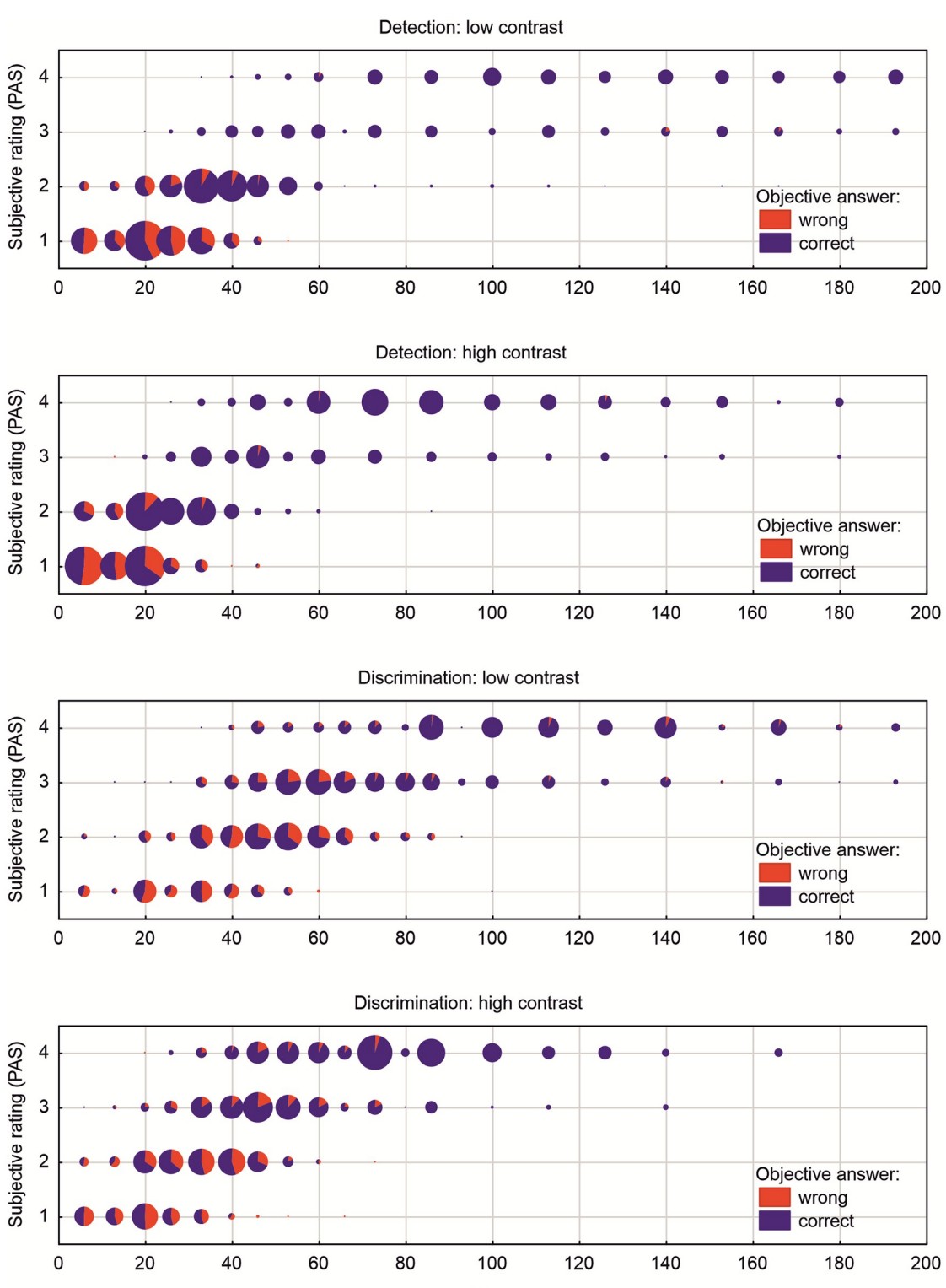

**Fig 3. Scatter-icon histograms for responses from all subjects.** The four graphs depict the variation of task (detection and discrimination) as well as variation of contrast (low and high). The abscissa shows the SOA in ms, truncated at 200 ms. The ordinate represents the four levels of the PAS scale, the subjective responses. The area of each dot represents the number of presentations per SOA with a response at the given PAS level. The pie chart in each dot depicts the ratio of correct (blue) and wrong (red) answers in the objective response (2-AFC).

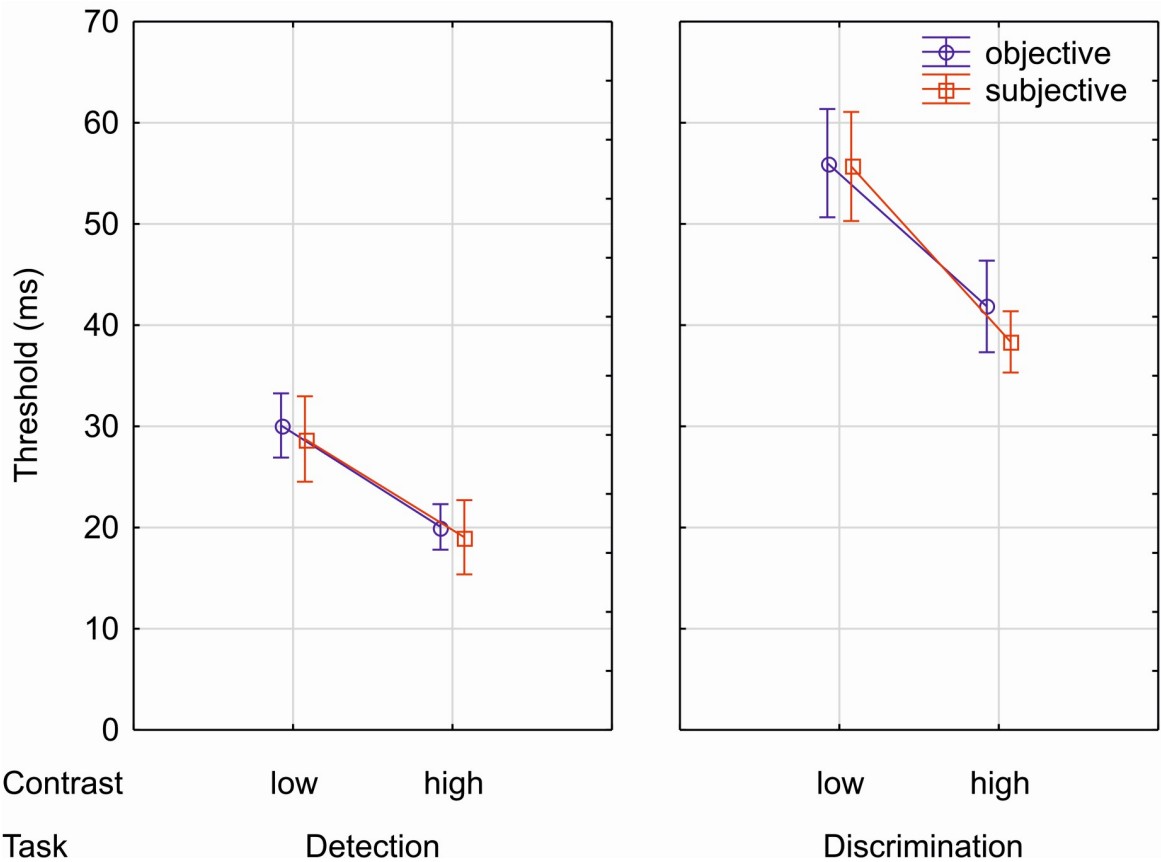

**Fig 4. Thresholds in ms for objective (blue) and subjective (red) responses.** Error bars depict 95% confidence intervals (within design). On the abscissa, the different conditions are shown.

discrimination task (words written in capital letters or small letters). Thresholds were higher for the low contrast condition compared to the high contrast condition. The significant interaction TASK x CONTRAST [$F(1,19) = 12.5$, $p = .002$, $\eta_p^2 = .397$] indicates that threshold differences between tasks were larger for the low than for the high contrast condition, although post-hoc tests revealed significant mean differences across all combinations of conditions (between tasks: at low contrast $p < .001$, at high contrast $p < .001$; between contrasts: detection task $p < .001$; discrimination task $p < .001$). The three interactions including the factor mode were not significant (for all tests F < 1.56, $p > .22$, $\eta_p^2 < .077$, S1 Table in S1 File). In fact, mean differences between the mode of responses (objective vs. subjective thresholds) were quite comparable. A Bayesian paired t-test for MODE (prior distribution: Cauchy 0.707) based on mean thresholds per subject collapsed across CONTRAST and TASK revealed a moderate evidence for $H_0$ (BF$_{01}$ = 3.045).

Following the suggestion of an anonymous referee, we provide threshold analyses in the supplement separately for event sequences, in which the target stimulus was presented either in interval 1 or in interval (S3 and S4 Tables in S1 File, S5 Fig in S1 File). Please note that these analyses should be cautiously interpreted due to the low number of trials in each event sequence. This analysis revealed lower estimated objective thresholds for the first interval with respect to second interval. Thresholds from subjective rating did not follow this tendency. Furthermore, subjective thresholds at intervals 1 and 2 were by and large in the middle of the respective objective thresholds.

### 3.3 Analysis of width

Similar to thresholds, width data (steepness of estimated psychometric functions) were subjected to a repeated-measures ANOVA with the factors TASK (absent/present and capital), CONTRAST (low and high) and MODE (objective and subjective). This analysis (Fig 5, S2 Table in S1 File) yielded significant main effects for the factors TASK [$F(1,19) = 17.7$, $p < .001$, $\eta_p^2 = .482$] and MODE [$F(1,19) = 49.0$, $p < .001$, $\eta_p^2 = .721$], whereas factor CONTRAST was not significant [$F(1,19) = 2.36$, $p = .14$, $\eta_p^2 = .11$]. The main effect TASK indicates that widths of the psychometric function for the detection task were lower (i.e. the slopes were steeper) compared to the discrimination task. Furthermore, the main effect MODE reflects lower widths for an objective response mode (i.e. the slopes were steeper) compared to a subjective one. Although the significant interaction CONTRAST x MODE [$F(1,19) = 20.4$, $p < .001$, $\eta_p^2 = .518$] is due to numerically larger width differences with respect to the response mode in the low contrast conditions compared to the high contrast conditions, widths obtained in the objective response mode were consistently smaller compared to widths obtained in the subjective response mode for both low ($p < .001$) and high contrast conditions ($p < .001$) according to post-hoc tests. Furthermore, post-hoc tests showed that for the subjective response mode widths were significantly lower for high than for low contrast ($p < .001$), whereas for the objective response mode widths of the contrast conditions did not significantly differ ($p > .08$). The remaining interactions were not significant (see S2 Table in S1 File).

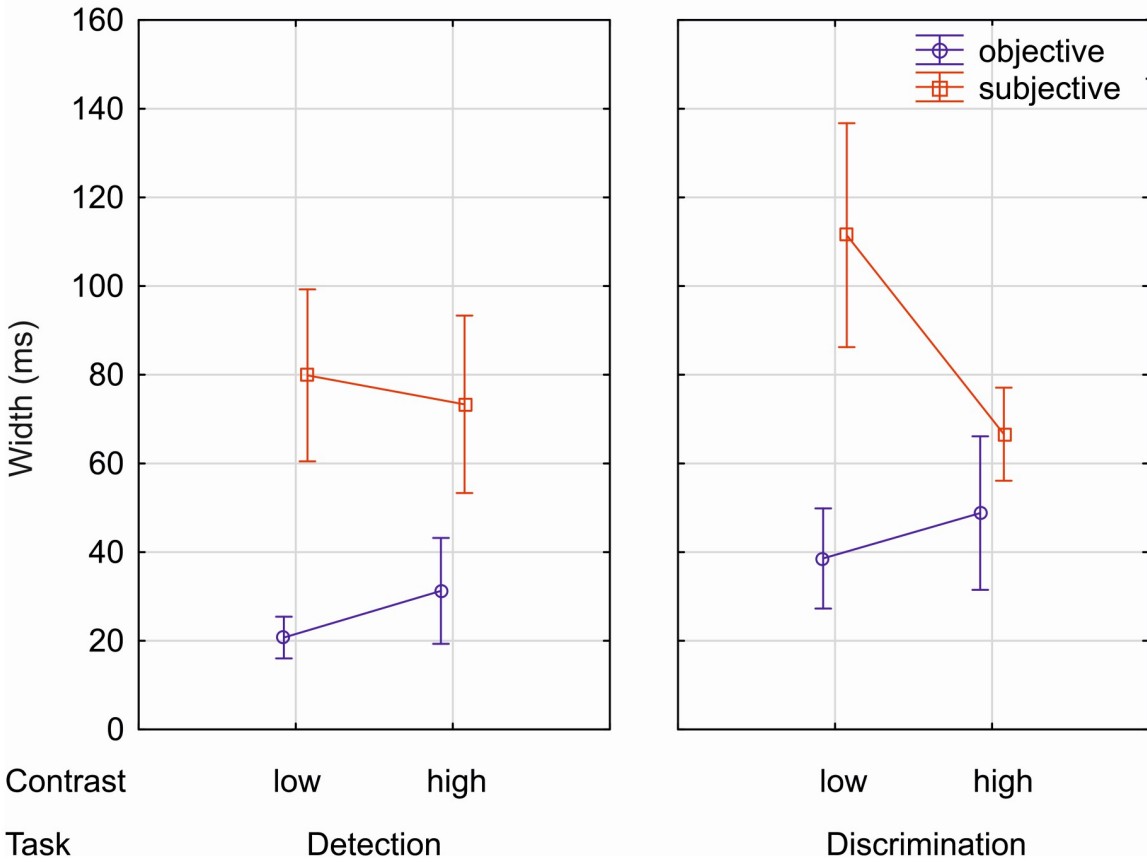

**Fig 5. Width of estimated psychometric functions (range 0.05–0.95, in ms) for objective (blue) and subjective (red) responses.** Error bars depict 95% confidence intervals (within design). On the abscissa, the different conditions are shown.

## 4. Discussion

The present study tested the relation between objective and subjective measurements of visual awareness by determining thresholds for visual word perception at an objective performance and a subjective judgment level using a temporal two-alternative forced choice task (temporal 2-AFC). Subjective ratings and objective performance measures were obtained for detection (stimulus presence) and discrimination (letter case) tasks at high and low stimulus contrast to assess awareness across different feature and difficulty levels. The present work served to discriminate between three different hypothetical scenarios for a relation between objective and subjective measures of awareness, outlined in the introduction section: Objective and subjective measures could capture (i) entirely different cognitive processes and states of awareness (independence of objective and subjective thresholds), (ii) related cognitive processes at different time scales reflecting different states of awareness (relatedness of objective and subjective thresholds with a constant lag) and (iii) the same cognitive processes reflecting the same states of awareness (zero lag between objective and subjective thresholds).

Results were clear-cut: Task and contrast significantly affected awareness thresholds: Thresholds were higher for the discrimination than for the detection task and for low than for high contrast. Furthermore, task differences were more pronounced at low than at high contrast. Most notably, mode of measurement (objective vs. subjective) did not affect obtained thresholds. In fact, mean thresholds in the different conditions were numerically almost identical for objective and subjective awareness measurements. Hence, this zero lag between objective and subjective thresholds indicates that both types of awareness measurements in principle similarly capture the content of visual awareness. Differences between objective and subjective measurements were found with regard to width of the psychometric function, which was larger for subjective than for objective awareness measurements, in particular for the low contrast condition. In addition, the width of the psychometric function was larger for the discrimination than for the detection task. This indicates that the transition from unaware to aware states proceeds more rapidly for objective compared to subjective measures and for the detection compared to the discrimination task. In the next section, we discuss the effects on thresholds before we move on to the effects on the width (or slopes) of the fitted psychometric functions.

Stimulus contrast affected awareness thresholds with higher thresholds in the low contrast condition compared to the high contrast condition. In line with the observation of slower reactions for low contrast stimuli [41], this finding indicates that low contrast stimuli need more time to be consolidated within the visual system before reaching awareness. Consistent with earlier work [5,35,39,40], detection thresholds (absent-present task) were lower than discrimination thresholds (capital task). This indicates that awareness of the presence or absence of a stimulus requires less time for visual consolidation and therefore emerges temporally earlier than awareness of specific stimulus features such as letter case. These task differences were more pronounced in the low than in the high contrast condition, because the low contrast resulted in a larger delay of the threshold for the discrimination task compared to the detection task. This interaction between task and contrast indicates that temporal dispersion of awareness of lower-level and higher-level visual feature is larger at low than at high contrast. Presumably, in the low contrast condition with impoverished physical information about stimulus features, stimuli have to be consolidated for a much longer time before individuals are aware of higher-level visual features compared to lower-level features such as experience of mere stimulus presence.

Most importantly to the purpose of the present study, this pattern of thresholds across tasks and contrasts was comparable for objective and subjective measurements of awareness. Hence,

both types of measurement converge on the temporal course at which participants are aware of stimulus features at different levels of complexity. This finding renders it likely that objective performance measures based on accuracy and subjective ratings of the visual experience can provide similar information on the feature-content of a percept. In line with the present observation of a zero lag between objective and subjective awareness thresholds, a recent study using constant mask stimulus SOAs in different experimental sessions found comparable d' sensitivity measures based upon objective performance and subjective PAS ratings [51]. However, the present observation of comparable subjective and objective thresholds contrasts with earlier proposals and empirical findings suggesting that subjective experience is independent from [31,32] or constantly lags behind objective performance [10,28]. We attribute this convergence of thresholds based on objective and subjective measures of awareness in the present study to three specific aspects of our experimental procedure: Firstly, our temporal 2-AFC task minimizes the influence of unconscious response tendencies on response accuracy, as outlined in the introduction section. Secondly, the participants in the present study were carefully trained in using the PAS ratings to report their subjective visual experience. This training might have enabled participants to rate their visual experience according to the PAS instruction without the need of extended visual processing to provide a rating of specific awareness level. In addition, for subjective ratings thresholds were defined in a task-dependent fashion (see Introduction). Finally, instructions in the study by Sandberg and colleagues [10] stressed response speed in addition to accuracy in the objective task, while there was no speed instruction for the PAS ratings as it is typical for ratings in general. In contrast, our instructions did not emphasize the speed of the response neither for objective performance nor for PAS ratings. It is possible that the speed instruction in earlier studies additionally induced responding based upon unconscious response priming in the objective task [34].

Despite these differences in the procedure of our study and earlier work, one might argue that the present observation of a convergence of subjective and objective thresholds reflect merely the fact that subjective and objective measures of awareness were collected within the same trial and are therefore not independent. This interpretation is however unlikely, because in previous research subjective and objective measures diverged, although they were also collected within the same trial [e.g., 10,28]. This shows that our observation of a converging pattern of subjective and objective thresholds is not trivial and merely an expression of a structural interdependence of measures. Although introduction of subjective ratings can in principle influence performance in the same [52] or in the next trial [44], we do not see how interactions between subjective and objective awareness tasks could induce a convergence of thresholds. Nevertheless, to specifically examine cross-task interactions, future studies could collect subjective and objective awareness measures in different trials (for a further discussion, see below).

Although the 2-AFC task minimizes biases from unconscious responding, biases originating from preferences for one response alternative might occur [53,54]. However, such response preferences get balanced, when analyses average across response alternatives. To reveal such response preference biases in the present data set, we analyzed thresholds in two subsets of the raw data, sorted by the interval in which the target stimulus was presented. Thresholds from objective responses presented in interval 1 were systematically lower compared to interval 2. This observation most likely reflects the tendency of observers to respond more frequently to have seen the target in interval 1 in case they did not see anything and were guessing. Interestingly, thresholds from subjective rating did not follow this tendency, but in 3 out of 4 conditions there was little variation of subjective thresholds for targets presented in interval 1 and interval 2. Furthermore, subjective thresholds at intervals 1 and 2 were by and large in the middle of the respective objective thresholds. Hence, the response bias related to interval 1 was

only present for objective, but not for subjective thresholds. We would like hasten to add that these analyses of thresholds separately for the event sequences should be cautiously interpreted due to the low number of trials. Obviously, when analyses were based on the intervals 1 and 2, this response preference with regard to interval 1 for objective performance gets balanced.

Slopes as indexed by the width parameter of the psychometric functions differed as a function of task, but also of response mode and contrast. Width was larger for the discrimination compared to the detection task, indicating a shallower slope for the higher-level task. This observation of steeper slopes for more basic perceptual features such as stimulus presence compared to more complex ones (letter case) is in line with earlier results using a similar temporal 2-AFC paradigm as in the present study [5,35], but contradicts findings in the context of the levels of processing approach [27,28,39]. Specifically in the study by Windey and colleagues [27], slopes for lower-level tasks were shallower than for higher-level tasks. As outlined in the introduction section, we assume that slopes of psychophysical functions and thus the transition from unawareness to awareness for a given feature are highly flexible and depend among other factors on the dimensional continuity of the features probed in the task [see also, 9]. In the studies testing the levels of processing approach, the low-level task frequently consisted of a hue judgment task, whereas the high-level task was a number comparison task. Dimensional continuity between blue and red hues, but also between different letter forms is more continuous compared with stimulus absence vs. presence or with integer number magnitude, which are both intrinsically discontinuous. It is conceivable that awareness of a feature on a continuous dimension emerges rather gradually compared with features on discontinuous dimensions, which give rise to sharp transitions between unaware and aware states. Most likely, for simple dichotomous perceptual features such as stimulus presence the state of awareness changes rapidly within a few milliseconds of processing, exhibiting a more discontinuous time course, where processing steps have to be more extended to produce changes from unawareness to awareness for more complex shape (or color) features. Interestingly, a similar more gradual transition from unawareness to awareness for complex vs. simple features has been suggested for somatosensory features associated with actions [55]. Possibly, a more gradual emergence of awareness for complex features might be a functional principle also in other domains than vision.

It should be noted at this place that the slope/width parameter of the psychometric function is independent of the threshold parameter. Hence, purely on mathematical grounds, the shallower slope and the higher threshold for the discrimination task compared to the detection task are two independent phenomena. If one wanted to account for the larger dispersion of the psychometric function in the discrimination task and considered a logarithmic SOA time scale, the slopes of the functions of the detection and discrimination tasks would be most likely more comparable. However, at present there are no convincing theoretical arguments for a preference of a logarithmic time scale over a linear one.

Slopes were also shallower for subjective ratings than for objective accuracy measurements in particular in the low contrast conditions. This difference in the slope of the psychometric function depending on the response mode is difficult to interpret, because complexity of responses and response classification differed between objective and subjective awareness measurements. In objective measurements, participants performed a 2-AFC task, and psychometric functions were fitted to the distribution of correct responses as a function of the SOA. In subjective measurements, participants used PAS to describe their perceptual experience ranging from complete unawareness, over a glimpse and partial awareness to a clear impression of the stimulus. Psychometric functions were fitted to the distribution of the four response categories as a function of the SOA. The different slopes for objective and subjective measurements might have two origins. Firstly, in the subjective measurements psychometric function

were fitted to the frequency distribution of the four discrete response categories compared to the continuous accuracy distribution in objective measurements, which could have affected their slope. Secondly, due to the discrete nature of the scale participants might have been somewhat cautious to move on to a PAS category indexing higher visibility with increasing SOA, in particular when the contrast is low and the percept is somewhat faded. Note that the different response categories of objective and subjective measurements of awareness potentially affect only the slope, but not the thresholds of the fitted psychometric functions. In line with this reasoning, our analyses relating detection and discrimination performance to PAS levels revealed above-chance task performance (a larger proportion of correct vs. wrong responses) predominantly for PAS ratings reporting subjective awareness of the feature in question (PAS level 2 "glimpse" → stimulus detection, PAS level 3 "partial awareness" → discrimination). As this analysis of frequency distributions does not depend on fitting of psychometric functions, it provides complementary evidence for a strong relation between subjective and objective awareness thresholds in the present study. Please also note that this analysis also validates our decision to determine the subjective threshold in the detection task at PAS level 1.5 and in the discrimination task at PAS level 2.5 (for a further discussion of this issue, see below).

We would like to emphasize at this place that the slope of a psychometric function (or its width) is a parameter influenced by several different factors, such as the form of observer's internal signal transducer, and the level of signal uncertainty [56]. Furthermore, non-stationarity caused by fluctuation of attention as well as perceptual learning will directly affect the slope. Up to now in psychophysics, there are only a few conditions, where variation in slope is systematically attributed to changes of the stimulus such as wavelength in visual detection threshold [57] or different pedestal levels in contrast detection and discrimination [58,59]. Although in consciousness research [e.g., 10,18], the slope parameter is frequently taken as index for the more gradual vs. more dichotomous emergence of visual awareness, psychophysical research suggests that the slope is affected by other factors as well, as outlined above, which renders an unequivocal interpretation of this parameter of the psychometric function difficult.

The present finding of comparable SOA thresholds as a function of task and contrast for objective and subjective assessments of awareness in our temporal 2-AFC task shows that objective performance measurements and subjective ratings of stimulus clarity can yield converging information about the content of a percept: The time course of consolidation of stimulus representations required for above-threshold performance for specific stimulus features closely corresponded with the time course of thresholds required for phenomenal experience related to this feature. Most importantly, the dispersion of SOA thresholds for subjective experiences of stimulus presence and of letter case features closely resembled the dispersion of SOA thresholds based on accurate performance in the low and high contrast conditions. This convergence of SOA thresholds suggests that both psychophysical and subjective approaches to awareness can provide converging and thus most likely veridical measures of awareness. Our results therefore reject popular criticisms of subjective and objective approaches, at least with regard to the present data set. The assumption that objective performance measures of awareness are mainly outcomes from unconscious processing or expressions of access consciousness and therefore no valid reflections of phenomenal experience [29] does not hold.

With regard to subjective awareness ratings, the present converging results validate subjective approaches of awareness by indicating that participants in the present study used the different categories of the PAS in a quite fine-grained and apparently valid fashion as instructed: For the detection task (stimulus presence), the SOA threshold in subjective ratings was determined at the PAS level 1.5, which lies between the level 1 "unseen" and the level 2 "glimpse of

something, but don't know what it was". Subjective SOA thresholds at this PAS level, which describes experience of stimulus presence, corresponded with the objective detection performance threshold (absent/present accuracy of 75% in 2-AFC task). As already outlined above, analysis of cumulative data relating task performance to levels of PAS ratings also validates this assumption. For the discrimination task (letter case), the SOA threshold in subjective ratings was determined at the PAS level 2.5, which lies between the level 2 "I saw a glimpse of something, but don't know what it was" and level 3 "I saw something, and I think I can determine what it was". Subjective SOA thresholds at this PAS level, which characterizes a visual experience of a specific stimulus, corresponded with the objective discrimination performance (letter case discrimination accuracy of 75% in 2-AFC task). This indicates that perceptual experiences intended to be reflected by PAS levels 2 vs. 3 nicely converge with feature-specific objective thresholds for experience of stimulus presence and experience of stimulus features, respectively. Again, this assumption was validated by the analysis of cumulative data relating task performance to levels of PAS ratings. Our results therefore reject the criticism that subjective ratings scales such as PAS do not appropriately characterize the visual experience of observers, because they are prone to biases [13,20,22,24,30].

Future replication studies involving larger samples and including even more manipulations of stimulus or task elements would be valuable in order to gain insights in the stability and generalizability of our zero lag observation. Furthermore, it would be desirable to directly test the assumption that the temporal 2-AFC used in the present study compared to the classical 2-AFC used in earlier studies [10,28] is a crucial factor for the convergence of subjective and objective thresholds across contrast and task conditions. Furthermore, as in the present study like in earlier studies [10,28] objective and subjective measures were collected within the same trial, future studies could also test whether the thresholds would still converge if the objective and subjective measures were collected in different blocks of the experiment.

The present findings have not only important implications of the measurement of awareness, but also for theories of consciousness. Our results indicate that the time course of visual consolidation giving rise to objective above-threshold performance, typically classified as an index of access consciousness coincides with the time course of the emergence of subjective perceptual experience, typically classified as reflection of phenomenal consciousness. This is intriguing insofar as access and phenomenal consciousness describe consciousness from a different perspective [1,2]. Furthermore, the generation and report of phenomenal experience is frequently supposed to require some extra processing costs [33] compared to processing leading to accurate performance. The temporal correspondence of thresholds for subjective experience and objective performance across tasks and contrasts indicates that the content and cognitive processes of access and phenomenal consciousness can be highly related, when probed in a manner, which minimizes biases from unconscious processing, as in the present study.

In conclusion, using a temporal 2-AFC task we found a comparable pattern of thresholds across tasks and contrasts for objective and subjective measurements of awareness. This finding suggests that objective performance measures based on accuracy and subjective ratings of the visual experience can provide similar information on the feature-content of a percept [see also, 51]. The observed similarity of thresholds validates both psychophysical and subjective approaches to awareness as converging and thus most likely veridical measures of awareness.

## Supporting information

**S1 File. Supplementary tables, figures and analyses.**
(PDF)

## Author Contributions

**Conceptualization:** Markus Kiefer, Thomas Kammer.

**Data curation:** Verena Frühauf, Thomas Kammer.

**Formal analysis:** Verena Frühauf, Thomas Kammer.

**Investigation:** Verena Frühauf, Thomas Kammer.

**Methodology:** Markus Kiefer, Thomas Kammer.

**Project administration:** Markus Kiefer, Thomas Kammer.

**Software:** Thomas Kammer.

**Supervision:** Markus Kiefer, Thomas Kammer.

**Visualization:** Thomas Kammer.

**Writing – original draft:** Markus Kiefer, Thomas Kammer.

**Writing – review & editing:** Markus Kiefer, Verena Frühauf, Thomas Kammer.

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
