## [Decision Letter · Decision Letter 0]

14 Apr 2023

PONE-D-23-03662Subjective and objective measures of visual awareness convergePLOS ONE

Dear Dr. Kiefer,

Thank you for submitting your manuscript to PLOS ONE. After careful consideration, we feel that it has merit but does not fully meet PLOS ONE’s publication criteria as it currently stands. Therefore, we invite you to submit a revised version of the manuscript that addresses the points raised during the review process. Editor comments: Two reviewers commented on you manuscript and I have read the manuscript myself. As you can see from their comments, both referees are generally positive though both raise some concerns about the conclusion finally drawn from the null effect of response mode (objective vs. subjective measures) on threshold, as it is based on a single experiment and may require replication across larger samples or a more effective experimental variation. They also have some reservations about the methodological approach to estimating thresholds from both measures of awareness and the analysis of the psychometric functions, suggesting a more detailed description in the methods section to clarify some aspects. Based on these comments and my own reading, I advice you preparing a revision of the manuscript and to provide more details and justifications in the methods section and clarify any vague statements. Additional empirical data would be fine (as it would increase the impact of this already fine work), though I would not strictly demand it.  

We look forward to receiving your revised manuscript.

Kind regards,

Michael B. Steinborn, PhD

Section Editor

PLOS ONE

Reviewers' comments:

Reviewer's Responses to Questions

**Comments to the Author**

1. Is the manuscript technically sound, and do the data support the conclusions?

Reviewer #1: Yes

Reviewer #2: Partly

2. Has the statistical analysis been performed appropriately and rigorously? 

Reviewer #1: Yes

Reviewer #2: I Don't Know

3. Have the authors made all data underlying the findings in their manuscript fully available?

Reviewer #1: Yes

Reviewer #2: No

4. Is the manuscript presented in an intelligible fashion and written in standard English?

Reviewer #1: Yes

Reviewer #2: Yes

5. Review Comments to the Author

Reviewer #1: The authors report one study, in which they investigated whether objective and subjective measures of visual awareness differ or not. They used two different tasks, a detection and a discrimination task and analysed objective (accuracy) and subjective (PAS) measures by means of psychometric analyses. The results show very similar detection and comparison thresholds for objective and subjective measures.

This study tackles a timely and important issue in consciousness research. The manuscript is well written and the results seem clear. My only concern with this study is that the results are dependent on how the thresholds are defined. For the objective accuracy this is clear, however, for the subjective case (PAS) it seems not so clear to me. The authors used the PAS levels 1.5 and 2.5 to determine the detection and the discrimination thresholds, respectively. I can intuitively follow this approach, yet I am wondering whether the PAS levels 2 and 3 wouldn't also make sense. Importantly, such thresholds would completely change the result pattern (higher thresholds for PAS than for accuracy) and also the conclusions of the study (in favour of a lag between subjective awareness and objective performance). I think the authors should clarify why these levels were chosen for the thresholds and why other levels are considered inappropriate.

Minor points

- Power analysis: the power analysis is based on d, even though the effect of interest is a 2 x 2 within-subject interaction. There is a paper by Langenberg et al. (2022), which highlights that the standard conversion of effect sizes (large effect: ηp2 =. 14 and dz = .80) is not appropriate for within-subject designs. For example, if one wants to find an interaction effect with ηp2 =. 14 (a large effect according to Cohen) the conversion to dz would yield dz = .40 (https://statsbyrandolph.psychologie.uni-bremen.de/Power/powerANOVA_gui). I am not sure how this problem could be solved (maybe new conventions need to be established), but it is probably better in repeated measures designs to have a desired ηp2 (instead of dz) for an a priori power analysis.

- Results: Please report p-values in APA conform format in all cases

- Please check the manuscript for typos

Literature

Langenberg, B., Janczyk, M., Koob, V., Kliegl, R., & Mayer, A. (2022). A tutorial on using the paired t test for power calculations in repeated measures ANOVA with interactions. Behavior Research Methods . https://doi.org/10.3758/s13428-022-01902-8

Reviewer #2: Review of PONE-D-23-03662

Subjective and objective measures of visual awareness converge

This manuscript reports the data of a single experiment in which participants had to detect masked words vs. identify the case of masked words with varying word-mask SOA. In addition to the task, also the contrast of the words was varied. Accuracy in the tasks served as an objective measure of visual awareness, and additional Perceptual Awareness Scale (PAS) ratings served as a subjective measure of awareness. The authors derived awareness thresholds from both measures. Both measures of awareness varied similarly with the task demands (detection vs. identification) and contrast. The central conclusion of the authors is based on a null effect of response mode (objective vs. subjective) on threshold – that both measures yield converging estimates of perceptual awareness.

In general, the approach to estimate and compare thresholds from both measures of awareness to be able to compare them directly seems quite smart and elegantEspecially with the present set of data, the conclusion that objective and subjective measures of awareness converge might be a bit overstated. However, I think that this study actually adds something useful to the debate of how to optimally measure subjective awareness (and avoid some common mistakes that have been outlined recently, e.g. Schmidt, 2015). Specifically, I do have several reservations about the manuscript, mostly pertaining to the methodology and to the conclusions.

1. While I find the fact that both thresholds vary to a very similar amount with task and contrast to be quite interesting, I am not as confident about the lack of an effect of “response mode” as it is basically a null effect. To be more convinced of this finding, I would like to see replication across larger samples and or more manipulations of the stimuli. This is especially in the light of previous findings with similar methodology that observed a lag between the measures, thus leading to very different conclusions (Sandberg et al., 2011). The authors suggestion that these differences might be due to the different task designs (e.g. the use of a temporal 2AFC task here) might be empirically validated.

2. Even though the authors already discuss this, it still is difficult for me to understand why the pronounced dissociation in the spread of the psychometric functions for both measures (interaction of response mode and contrast) would not invalidate the conclusion that both measures reflect converging measures of awareness. Especially the author’s point ”psychometric functions were fitted to the frequency distribution of the four discrete response categories compared to the continuous accuracy distribution in objective measurements, which could have affected their slope” (p. 22) is too vague. If this is true, I would suspect that this might also have affected threshold measurements or at least their stability? I think there should be some reference or proof of concept that validates this specific estimation procedure.

3. Relatedly, the authors suggest that an important factor for the diverging results might be the employment of a temporal 2AFC task because it is “free of bias”. Actually this is not completely true: the data from a 2AFC task still may include biases (e.g., preferring one response or interval over the other) but this gets balanced out in averaging across the data in which the correct target is presented in the first or the second interval. Crucially, the presence of such a “conceiled” bias will affect (typically artificially enlarge) estimates of the spread of the psychometric function. It is therefore advisable to analyse data from both trial types (target in first interval and target in second interval) separately – or jointly, by taking the trial types into account (cf. Vorberg and Ulrich, 2009, Ulrich, 2010)

4. One potential complicating fact is that both measures of awareness were assessed in the same trials (and always in the same order). I suspect that PAS ratings might at least to some degree be highly biased by the “objective” responses given directly before, e.g., by their ease, or by some monitoring process. I wonder – and this is an additional experiment that I really would like to see – whether the thresholds would still converge as well if the objective and subjective measures were collected in different blocks of the experiment.

5. Relatedly, is it warranted to enter thresholds derived from the two different measures (here_ response modes) in the same ANOVA? Especially because these were collected in the same trials and thus might not be independent.

6. Even though I found the manuscript sometimes a bit lengthy and slightly repetitive, in other parts (especially the methods section) it seems to lack detail. For example, I did not find the rationale why three runs of the adaptive SOAs were used – and which color in Fig. refers to which type of run? Why are the runs depicted in Fig. 2 of different lengths? Why did the authors decide to use an adaptive procedure rather than the method of constant stimuli which would provide equal numbers of trials for the different SOA durations? The word stimuli (in Fig. 1 it seems it is actually rather pseudowords?) should be described in more detail. Maybe most crucially, how was the training of the PAS actually administered? What was the criterion for using the scale correctly? Depending on the exact training procedure, such a training might very strongly prompt the participants to base their subjective ratings on objective performance (or at least the ease with which the decisions could be made), or maybe to form an association between the PAS categories and the SOAs.

7. On page 9 it is stated that a larger width of the psychometric function indicates a more gradual transition from unconscious to conscious. Could it also just indicate an abrupt transition which however varies on a trial to trial basis? Averaging across many such trials might also lead to a flattening of the psychometric function

8. I think the full ANOVA results should be reported in the main text rather than as supplementary material, as all possible interactions are relevant to the interpretation of the results.

9. I hope I did not overlook it but the osf repository only seems to include the aggregated data (parameters of the fitted function). Should not also raw data be included according to the PLOS standards?

10. Some of the highly relevant empirical background could be described in more detail in the Introduction. For example, it would help if the details of the Sandberg study (and its differences to the present study) would be described more clearly. Being not an expert in the field of perceptual awareness, it was difficult for me to figure out what exactly is new in this study compared to previous ones. Given the more general scope of PLOS, I feel that many readers might encounter a similar problem.

Sandberg, K., Bibby, B. M., Timmermans, B., Cleeremans, A., & Overgaard, M. (2011). Measuring consciousness: task accuracy and awareness as sigmoid functions of stimulus duration. Consciousness and cognition, 20(4), 1659-1675.

Schmidt, T. (2015). Invisible stimuli, implicit thresholds: Why invisibility judgments cannot be interpreted in isolation. Advances in Cognitive Psychology, 11(2), 31.

Ulrich, R., & Vorberg, D. (2009). Estimating the difference limen in 2AFC tasks: Pitfalls and improved estimators. Attention, Perception, & Psychophysics, 71(6), 1219-1227.

Ulrich, R. (2010). DLs in reminder and 2AFC tasks: Data and models. Attention, Perception, & Psychophysics, 72(4), 1179-1198.

6. PLOS authors have the option to publish the peer review history of their article (what does this mean?). If published, this will include your full peer review and any attached files.

Reviewer #1: No

Reviewer #2: No

---

## [Author Response · Author response to Decision Letter 0]

14 Jul 2023

PONE-D-23-03662

Subjective and objective measures of visual awareness converge

PLOS ONE

Responses to the Reviewers

We would like to thank the reviewers for their positive evaluation of our manuscript and the thoughtful and stimulating comments. When preparing the revision, we have carefully taken all comments into account and have changed the manuscript accordingly. We feel that these suggestions have substantially improved the quality of the manuscript. All changes are marked in the manuscript in yellow color.

Reviewer #1

The authors report one study, in which they investigated whether objective and subjective measures of visual awareness differ or not. They used two different tasks, a detection and a discrimination task and analysed objective (accuracy) and subjective (PAS) measures by means of psychometric analyses. The results show very similar detection and comparison thresholds for objective and subjective measures.

This study tackles a timely and important issue in consciousness research. The manuscript is well written and the results seem clear. My only concern with this study is that the results are dependent on how the thresholds are defined. For the objective accuracy this is clear, however, for the subjective case (PAS) it seems not so clear to me. The authors used the PAS levels 1.5 and 2.5 to determine the detection and the discrimination thresholds, respectively. I can intuitively follow this approach, yet I am wondering whether the PAS levels 2 and 3 wouldn't also make sense. Importantly, such thresholds would completely change the result pattern (higher thresholds for PAS than for accuracy) and also the conclusions of the study (in favour of a lag between subjective awareness and objective performance). I think the authors should clarify why these levels were chosen for the thresholds and why other levels are considered inappropriate.

Response: 

Thank you for rising the important point of threshold definition. Indeed, it is common practice to define the threshold within the psychometric function on its half maximal performance value (0.75 in case of 2AFC), i.e. at the transition point between uncertainty and certainty. We aimed to transfer the definition of threshold as half maximal performance into the metrics from subjective responses (PAS). To consider uncertainty in the responses to PAS we did not define thresholds at a given PAS level but always on the transition between two levels, i.e. 1.5 as transition between "I've seen nothing" (PAS level 1) and "I saw a glimpse of something, but don’t know what it was" (PAS level 2) in case of detection tasks, and 2.5 as transition between "glimpse" (PAS level 2) and "I saw something, and I think I can determine what it was" (PAS level 3). We added this explanation in the Introduction. It now reads: 

"In analogy to the objective threshold definition of 0.75, i.e. the transition between pure guessing (0.5) and correct response (1.0) in the 2-AFC task, we therefore defined the subjective thresholds from the fitted psychophysical functions in the stimulus detection task at PAS level 1.5 (transition between "nothing" and "glimpse") and in the stimulus discrimination task at PAS level 2.5 (transition between "glimpse" and "I saw something"). "

Furthermore, we added our new analyses relating detection and discrimination performance to PAS levels (see, p. 18 and Figure 3). This analysis revealed above-chance task performance predominantly for PAS ratings reporting subjective awareness of the feature in question (PAS level 2 “glimpse” � stimulus detection, PAS level 3 “partial awareness” � discrimination). As this analysis of frequency distributions does not depend on fitting of psychometric functions, it provides complementary evidence for a strong relation between subjective and objective awareness thresholds in the present study. Please also note that this analysis also validates our decision to determine the subjective threshold in the detection task at PAS level 1.5 and in the discrimination task at PAS level 2.5. This analysis clearly demonstrates that for detection task a meaningful threshold for subjective responses indeed is observed at PAS 1.5, whereas a meaningful threshold for discrimination is observed at PAS 2.5. We added a discussion of this new analysis on p. 27.

Minor points

- Power analysis: the power analysis is based on d, even though the effect of interest is a 2 x 2 within-subject interaction. There is a paper by Langenberg et al. (2022), which highlights that the standard conversion of effect sizes (large effect: ηp2 =. 14 and dz = .80) is not appropriate for within-subject designs. For example, if one wants to find an interaction effect with ηp2 =. 14 (a large effect according to Cohen) the conversion to dz would yield dz = .40 (https://statsbyrandolph.psychologie.uni-bremen.de/Power/powerANOVA_gui). I am not sure how this problem could be solved (maybe new conventions need to be established), but it is probably better in repeated measures designs to have a desired ηp2 (instead of dz) for an a priori power analysis.

- Results: Please report p-values in APA conform format in all cases

- Please check the manuscript for typos

Literature

Langenberg, B., Janczyk, M., Koob, V., Kliegl, R., & Mayer, A. (2022). A tutorial on using the paired t test for power calculations in repeated measures ANOVA with interactions. Behavior Research Methods . https://doi.org/10.3758/s13428-022-01902-8

Response:

(i) Thank you for commenting on the effect size measure, on which the power analysis is based, and for drawing our attention to the interesting paper by Langenberg et al. (2022). Reading the Langenberg et al. (2022) entirely convinced us that ηp2 is the more appropriate effect size measure for interaction effects in within-subject designs and d is not appropriate. We therefore now report ηp2 in the revised manuscript, but hasten to add at this place that the type of effect size measure cannot be changed in the preregistration. Please note that we did not use the labels do “convert” ηp2 to d/f, but used Gpower to perform the conversion. We just reported effect size in terms of d/f, because this measure was entered in power analysis. Motivated by this reviewer’s comment, we have now checked that the conversion implemented in Gpower corresponds to the conversion formula described in Langenberg et al. (2022). The effect size was taken from an unpublished pilot study yielding a ηp2 = 0.39, which approximately corresponds to a d/f = 0.80. This information is now provided in detail on p. 12 in the methods section.

(ii) Thank you for drawing our attention to a consistent reporting of p values according to APA guidelines. We changed the format of the report of p values and other statistics, if deviating from the APA style.

(iii) We checked the manuscript carefully for typos. 

Reviewer #2

This manuscript reports the data of a single experiment in which participants had to detect masked words vs. identify the case of masked words with varying word-mask SOA. In addition to the task, also the contrast of the words was varied. Accuracy in the tasks served as an objective measure of visual awareness, and additional Perceptual Awareness Scale (PAS) ratings served as a subjective measure of awareness. The authors derived awareness thresholds from both measures. Both measures of awareness varied similarly with the task demands (detection vs. identification) and contrast. The central conclusion of the authors is based on a null effect of response mode (objective vs. subjective) on threshold – that both measures yield converging estimates of perceptual awareness.

In general, the approach to estimate and compare thresholds from both measures of awareness to be able to compare them directly seems quite smart and elegant. Especially with the present set of data, the conclusion that objective and subjective measures of awareness converge might be a bit overstated. However, I think that this study actually adds something useful to the debate of how to optimally measure subjective awareness (and avoid some common mistakes that have been outlined recently, e.g. Schmidt, 2015). Specifically, I do have several reservations about the manuscript, mostly pertaining to the methodology and to the conclusions.

1. While I find the fact that both thresholds vary to a very similar amount with task and contrast to be quite interesting, I am not as confident about the lack of an effect of “response mode” as it is basically a null effect. To be more convinced of this finding, I would like to see replication across larger samples and or more manipulations of the stimuli. This is especially in the light of previous findings with similar methodology that observed a lag between the measures, thus leading to very different conclusions (Sandberg et al., 2011). The authors suggestion that these differences might be due to the different task designs (e.g. the use of a temporal 2AFC task here) might be empirically validated.

Response: 

We thank this reviewer for his or her positive evaluation of our findings and for describing our result pattern of a comparable variation of thresholds as a function of task and contrast to be quite interesting. We also agree with this reviewer that a replication across larger samples and even more manipulations would be valuable in order to gain insights in the stability and generalizability of our zero lag observation. Furthermore, it would be desirable to directly test the assumption that the task design (temporal 2AFC as in the present study vs. classical 2AFC as in the Sandberg et al. 2011 study) is a crucial factor for the presence or absence for a lag between subjective and objective thresholds. We entirely agree with this reviewer that our research stimulates a multitude of new research question concerning the measurement of awareness, which can and should be empirically addressed. However, the questions raised by this reviewer here in his or her point 1 or below in point 4 cannot be answered by one single new experiment, but need to be addressed in a series of several new experiments. While we fully share this reviewer’s interest in new data to address all these question, we think that inclusion of all suggested new experiments would render the present manuscript overly lengthy. Moreover, the appropriate measurement of awareness is an issue, which cannot be fully addressed in one article due to its complexity at the theoretical and empirical level. We therefore prefer to report only the present study as it is. In order to highlight that the present study should be considered as a starting point for a new line of research relating subjective and objective measures of awareness across different tasks and conditions, we now outline in the discussion section the new research questions raised by this reviewer, which could be addressed in future studies (p. 30). 

2. Even though the authors already discuss this, it still is difficult for me to understand why the pronounced dissociation in the spread of the psychometric functions for both measures (interaction of response mode and contrast) would not invalidate the conclusion that both measures reflect converging measures of awareness. Especially the author’s point ”psychometric functions were fitted to the frequency distribution of the four discrete response categories compared to the continuous accuracy distribution in objective measurements, which could have affected their slope” (p. 22) is too vague. If this is true, I would suspect that this might also have affected threshold measurements or at least their stability? I think there should be some reference or proof of concept that validates this specific estimation procedure.

Response: 

As we already mentioned in the discussion, the dissociation between thresholds and slopes in the psychometric functions is not trivial. We now expand the discussion of the slopes on p. 28 in the manuscript. In contrast to threshold, the slope of a psychometric function (or its width) is a parameter influenced by several different factors, such as the form of observer's internal signal transducer, and the level of signal uncertainty (Wallis et al. 2013). Furthermore, non-stationarity caused by fluctuation of attention as well as perceptual learning will directly affect the slope. Up to now in psychophysics, there are only a few conditions, where variation in slope is systematically attributed to changes of the stimulus (e.g. wavelength in visual detection threshold, Maloney 1990, or different pedestal levels in contrast detection and discrimination, Bird et al. 2002, Meese et al. 2006). We compared objective and subjective threshold data based on completely different scales. As we addressed in the introduction, the observed concordance between objective and subjective thresholds was not the leading hypothesis but the third of three different theoretical alternatives. Since the two scales have different origins (relative amount of correct responses vs distribution of graded responses from PAS scale with four levels) we cannot figure out a model where the psychometric functions of the two measurements result in concordant slopes. Turning the argument of the reviewer upside-down, we think that the concordance of subjective and objective thresholds with discordant slopes supports the importance of the finding since it is extremely implausible that discordant slopes would for some reason result in concordant thresholds. Furthermore, our new analyses relating detection and discrimination performance to PAS levels (see, p. 18 and Figure 3) revealed above-chance task performance predominantly for PAS ratings reporting subjective awareness of the feature in question (PAS level 2 “glimpse” � stimulus detection, PAS level 3 “partial awareness” � discrimination). As this analysis of frequency distributions does not depend on fitting of psychometric functions, it provides complementary evidence for a strong relation between subjective and objective awareness thresholds in the present study. Please also note that this analysis also validates our decision to determine the subjective threshold in the detection task at PAS level 1.5 and in the discrimination task at PAS level 2.5. We added a discussion of this new analysis on p. 27.

3. Relatedly, the authors suggest that an important factor for the diverging results might be the employment of a temporal 2AFC task because it is “free of bias”. Actually this is not completely true: the data from a 2AFC task still may include biases (e.g., preferring one response or interval over the other) but this gets balanced out in averaging across the data in which the correct target is presented in the first or the second interval. Crucially, the presence of such a “conceiled” bias will affect (typically artificially enlarge) estimates of the spread of the psychometric function. It is therefore advisable to analyse data from both trial types (target in first interval and target in second interval) separately – or jointly, by taking the trial types into account (cf. Vorberg and Ulrich, 2009, Ulrich, 2010)

Response: 

We thank this reviewer for raising the issue of a possible bias in temporal 2AFC tasks. First of all, we emphasize that we never have claimed that the 2AFC is “free of bias”. We have just noted that (i) the temporal 2AFC minimizes biases from responding based upon unconscious response priming and (ii) specifically with regard to the detection task (stimulus presence vs. absence) the temporal 2-AFC task involves a true two-alternative choice reaction (“Is the critical feature in the first or in the second interval?") irrespective of the complexity of the probed feature (stimulus presence vs. letter case). We entirely agree with this reviewer that thresholds determined in a (temporal) 2AFC should not be influenced by putative response preferences, since presentation of targets in the two presentation intervals is randomized and strictly balanced. In this sense, the complete dataset with combined analysis across response alternatives (here: interval 1 and 2) indeed is free of bias. To emphasize this aspect, we now explicitly state that in the temporal 2AFC response biases get balanced by averaging across response alternatives (p. 25). Thank you for suggesting the articles addressing order effects in a temporal 2AFC task (Ulrich and Vorberg, 2009, Ulrich, 2010). We read these publications with great interest. Ulrich and Vorberg describe 2AFC experiments including a constant stimulus and comparison stimuli of varying intensity in order to determine the just noticeable difference. In such a psychophysical experiment, participants have to indicate the interval, in which the more intense stimulus is presented. Ulrich and Vorberg further consider a situation, in which the comparison stimulus has always a larger or equal intensity than the constant. As a consequence, responses to comparison stimuli with lower intensity than the comparison could not be collected, which could induce several biases. Please note that this experimental situation, to which the work by Ulrich and colleagues refers, is substantially different from our version of the temporal 2AFC. In our 2AFC task, two fixed stimulus alternatives (word vs. nothing, uppercase word vs. lowercase word) were presented in intervals 1 and 2. Furthermore, in principle, the critical SOA variation, which determines stimulus visibility, was equally applied to both intervals. Of course, in the detection task, the SOA variation could only be applied to the interval with a present stimulus. Given these fundamental differences, we do not see how the criticism and correction procedures presented in the articles by Ulrich and Vorberg apply to our study.

Nevertheless, as suggested by this reviewer we reanalyzed our data separately by the interval of presentation. Estimation of thresholds on the basis of 20-40 observations only is problematic, but in most cases we obtained a meaningful threshold from sigmoid fitting. It turned out that, as the reviewer postulated, with lower SOAs in the objective responses there is a bias towards the first interval, so that estimated thresholds were lower for the first interval with respect to second interval. Of course, this does not indicate that objective thresholds are truly lower if the target is flashed in the first interval, but most likely reflects the tendency of observers to respond more frequently to have seen the target in interval 1 in case they did not see anything and were guessing. Again, this tendency towards reporting to have seen the target in interval 1 underlines the importance to combine performance accuracy across response alternatives. Interestingly, thresholds from subjective rating did not follow this tendency, but in 3 out of 4 conditions there was little variation of subjective thresholds for targets presented in interval 1 and interval 2. Furthermore, subjective thresholds at intervals 1 and 2 were by and large in the middle of the respective objective thresholds. Hence, the response bias related to interval 1 was only present for objective, but not for subjective thresholds. We would like hasten to add that these analyses of thresholds separately for the event sequences should be cautiously interpreted due to the low number of trials. Obviously, when analyses were based on the intervals 1 and 2, this response preference with regard to interval 1 for objective performance gets balanced. In our view, this nicely corroborates the robustness of the result that objective and subjective thresholds converge. We added this sub-analysis to the supplement and discuss the results in the main text on p. 25.

4. One potential complicating fact is that both measures of awareness were assessed in the same trials (and always in the same order). I suspect that PAS ratings might at least to some degree be highly biased by the “objective” responses given directly before, e.g., by their ease, or by some monitoring process. I wonder – and this is an additional experiment that I really would like to see – whether the thresholds would still converge as well if the objective and subjective measures were collected in different blocks of the experiment.

Response: 

We thank this reviewer for the suggestion to collect objective and subjective measures in different blocks of trials and to see whether this changes the pattern of thresholds. It is indeed an interesting idea that the response in the “objective” task may bias the PAS ratings. To our knowledge, up to now this hypothesis has not been tested so far, perhaps because a collection of objective and subjective thresholds in different experimental blocks is inconsistent with the argument of some scholars to prefer subjective ratings of awareness over objective psychophysical measurements, in order to capture trial-wise fluctuations of awareness (e..g, Lähteenmäki, M., Hyönä, J., Koivisto, M., & Nummenmaa, L. (2015). Affective processing requires awareness. Journal of Experimental Psychology: General, 144(2), 339-365. doi: 10.1037/xge0000040.). Furthermore, there might be sequence or training effects across blocks of trials, so that subjective and objective measurements collected in different blocks of trials are not comparable. Nevertheless, we find this reviewer’s suggestion interesting and worth to pursue in a new study. As we have outlined in detail already in response to this reviewer’s point 1, the appropriate measurement of awareness is an issue, which cannot be fully addressed in one article due to its complexity at the theoretical and empirical level. We therefore prefer to report only the present study as it is. However, in order to stimulate future work in this direction, we mention this point as a potential target in future work (p. 30).

5. Relatedly, is it warranted to enter thresholds derived from the two different measures (here_ response modes) in the same ANOVA? Especially because these were collected in the same trials and thus might not be independent.

Response: 

Thank you for raising this important question of entering subjective and objective measures in the same repeated-measures ANOVA by including the factor response mode. We think that independence of these measure is warranted because they can in principle vary independently from each other. Please note that in previous research subjective and objective measures diverged, although they were also collected within the same trial (Jimenez, M., Villalba-Garcia, C., Luna, D., Hinojosa, J. A., & Montoro, P. R. (2019). The nature of visual awareness at stimulus energy and feature levels: A backward masking study. Attention Perception & Psychophysics, 81(6), 1926-1943. doi: 10.3758/s13414-019-01732-5; Sandberg, K., Bibby, B. M., Timmermans, B., Cleeremans, A., & Overgaard, M. (2011). Measuring consciousness: Task accuracy and awareness as sigmoid functions of stimulus duration. Consciousness and Cognition, 20(4), 1659-1675. doi: 10.1016/j.concog.2011.09.002.). This also shows that our observation of a converging pattern of subjective and objective thresholds is not trivial and merely an expression of a structural interdependence of measures. We now added a short discussion of this issue on p. 24. We would also like to emphasize that collecting two measures within one trial and enter them in a common repeated-measures ANOVA is an essential feature in dual task research, in which participants deliver two responses in quick succession (Hommel, B., & Eglau, B. (2002). Control of stimulus-response translation in dual-task performance. Psychological Research, 66(4), 260-273. doi: 10.1007/s00426-002-0100-y; Koch, I., Poljac, E., Müller, H., & Kiesel, A. (2018). Cognitive structure, flexibility, and plasticity in human multitasking - An integrative review of dual-task and task-switching research. Psychological Bulletin, 144(6), 557-583. doi: 10.1037/bul0000144; Strobach, T. (2020). The dual-task practice advantage: Empirical evidence and cognitive mechanisms. Psychonomic Bulletin & Review, 27(1), 3-14. doi: 10.3758/s13423-019-01619-4).

6. Even though I found the manuscript sometimes a bit lengthy and slightly repetitive, in other parts (especially the methods section) it seems to lack detail. For example, I did not find the rationale why three runs of the adaptive SOAs were used – and which color in Fig. refers to which type of run? Why are the runs depicted in Fig. 2 of different lengths? Why did the authors decide to use an adaptive procedure rather than the method of constant stimuli which would provide equal numbers of trials for the different SOA durations? The word stimuli (in Fig. 1 it seems it is actually rather pseudowords?) should be described in more detail. Maybe most crucially, how was the training of the PAS actually administered? What was the criterion for using the scale correctly? Depending on the exact training procedure, such a training might very strongly prompt the participants to base their subjective ratings on objective performance (or at least the ease with which the decisions could be made), or maybe to form an association between the PAS categories and the SOAs.

Response: 

The reviewer addresses the critical balance between details to be reported from the scientific scope of replicability on the one hand and the length of a manuscript on the other hand. We agree that it's our responsibility to keep this balance. Therefore, we decided to put some issues into a supplement. Detailed responses to the raised questions can be found below: 

(i) " I did not find the rationale why three runs of the adaptive SOAs were used ": 

In case of restricted stimulus material, adaptive staircases have the advantage to sample data at threshold transitions of psychometric functions, helping to optimize information gained from the presented material. In our experience two or more independent staircases help to sample data across the whole range of a psychometric function, since they could start at different levels. In the present study it was a challenge to sample for two different psychometric functions (objective performance and subjective ratings) within one run, without knowing whether thresholds for the two functions differ or not. From pilot experiments we realized that this would be possible by at least two staircases controlled by both the objective and the subjective responses. Piloting work also indicated that it data sampling is improved, if one staircase is controlled by subjective ratings. We therefore added a third independent staircase in order to start at three different SOAs: a subthreshold level (13 ms, controlled by objective responses) as well as a putative suprathreshold level (47 ms, controlled by objective responses), and a clearly suprathreshold level (80 ms, controlled by subjective responses) in order to guarantee a sampling of subjective responses with PAS responses of 3 and 4. Subjective data in this range are a prerequisite for a successful estimation of a psychometric function. Please notice that prior to the experiment we did not know, whether subjective thresholds were in the same SOA range as the objective thresholds, as explicitly indicated in our hypotheses. Fortunately, with the run of all three staircases we sampled responses for both, the objective as well as the subjective ones, across the entire range of the psychometric function.

The passage in the methods section now reads (p. 14, lines 309-315): " Three randomly interleaved simple staircases (2 correct answers: SOA down; 1 incorrect answer: SOA up) were used to adaptively vary SOA over the 76 trials. Two of them were controlled by objective responses. They started at 20 ms and 40 ms, with a step size of 13 ms each. The third staircase covered longer SOAs and was controlled by subjective responses. The "correct" criterion was a response of 4, and response below 4 was considered as "wrong". This staircase started at 73 ms with a step size of 26.7 ms (Fig. 2)."

(ii) " and which color in Fig. refers to which type of run?"

Three staircases are plotted in three different colors. In the figure caption, now the start values of each staircase is explicitly stated.

(iii) " Why are the runs depicted in Fig. 2 of different lengths? "

As stated in the methods section, the three staircases were randomly interleaved, thus randomly chosen from trial to trial. Since trial number was fixed, true random choice must result in different and varying length of the three different staircases.

(iv) " Why did the authors decide to use an adaptive procedure rather than the method of constant stimuli which would provide equal numbers of trials for the different SOA durations?"

As already explained in detail in our response to point (i), in case of restricted stimulus material adaptive staircases have the advantage to sample data at threshold transitions of psychometric functions, helping to optimize information gained from the presented material.

(v) " The word stimuli (in Fig. 1 it seems it is actually rather pseudowords?) should be described in more detail." 

The three words depicted in Figure 1 are words from true English vocabulary, not pseudowords. Please notice that in the experiment only German words have been flashed. The five word lists used were identical to the word lists presented in the study reported in Kiefer and Kammer (2017). 

(vi) " Maybe most crucially, how was the training of the PAS actually administered? What was the criterion for using the scale correctly? Depending on the exact training procedure, such a training might very strongly prompt the participants to base their subjective ratings on objective performance (or at least the ease with which the decisions could be made), or maybe to form an association between the PAS categories and the SOAs."

We already addressed the issue of coupling objective and subjective response in the answer to question 4. The PAS training was exactly applied as described in the methods section (pp. 14-15, lines 319-332). The criteria for the different levels of the PAS scale are clearly documented in the passage: "PAS 4: ability to read the full word (full visibility), PAS 3: awareness of only some letters without the full word, PAS 2: glimpse." The training of the PAS was not combined with any training on the objective response (p. 15, lines 331-332). Please notice that we did not give feedback at all in the experiment. But indeed a glimpse is sufficient to "easily" respond to the detection task, similar to the perception of some capital letters in case of the discrimination task. This is a simple consequence of the nature of the task, that does not require any explicit training. 

7. On page 9 it is stated that a larger width of the psychometric function indicates a more gradual transition from unconscious to conscious. Could it also just indicate an abrupt transition which however varies on a trial to trial basis? Averaging across many such trials might also lead to a flattening of the psychometric function

Response: 

We agree with this reviewer that smaller steepness of the psychometric function might also result from an abrupt, but temporally variable transition. Note, however, that a potential substantial presence of intermediate PAS ratings (PAS levels 2 and 3), which indicate states of partial awareness, renders the interpretation in terms of variable dichotomous transitions unlikely. In the revised manuscript, we discuss this alternative interpretation of the slope of the psychometric function on p. 28 and also moderate the wording (“is taken to index”), when introducing the standard interpretation of the slope. Please also note that the interpretation of the slope as an index of transition from unawareness to awareness has been introduced by others and has been frequently used since then in consciousness research (Koch, C., & Preuschoff, K. (2007). Betting the house on consciousness. Nature Neuroscience, 10(2), 140-141. doi: 10.1038/Nn0207-140; Sandberg, K., Bibby, B. M., Timmermans, B., Cleeremans, A., & Overgaard, M. (2011). Measuring consciousness: Task accuracy and awareness as sigmoid functions of stimulus duration. Consciousness and Cognition, 20(4), 1659-1675. doi: 10.1016/j.concog.2011.09.002). We are fully aware of the multitude of factors influencing the slope and discuss slope effects in a balanced fashion (p. 28, lines 645-656). 

8. I think the full ANOVA results should be reported in the main text rather than as supplementary material, as all possible interactions are relevant to the interpretation of the results.

Response: 

We agree that full ANOVA results have to be provided in the paper. In contrast to the reviewer we think that not all possible interactions are relevant. If the reader is interested in a particular result she or he might take a look into the supplement.

9. I hope I did not overlook it but the osf repository only seems to include the aggregated data (parameters of the fitted function). Should not also raw data be included according to the PLOS standards?

Response: 

We now provide raw data in addition to the aggregated data.

10. Some of the highly relevant empirical background could be described in more detail in the Introduction. For example, it would help if the details of the Sandberg study (and its differences to the present study) would be described more clearly. Being not an expert in the field of perceptual awareness, it was difficult for me to figure out what exactly is new in this study compared to previous ones. Given the more general scope of PLOS, I feel that many readers might encounter a similar problem.

Response: 

As suggested, we have expanded the description of the Sandberg study and highlight the procedural difference to the present study in more detail (in the introduction on p. 5 & 6, and in the discussion on p. 24). There are the following major procedural differences: Firstly, our temporal 2AFC task minimizes the influence of unconscious response tendencies on response accuracy, as outlined in the introduction section. Secondly, the participants in the present study were carefully trained in using the PAS ratings to report their subjective visual experience. This training might have enabled participants to rate their visual experience according to the PAS instruction without the need of extended visual processing to provide a rating of a specific awareness level. In addition, for subjective ratings threshold were defined in a task-dependent fashion (see Introduction). Finally, instructions in the study by Sandberg and colleagues (2011) stressed response speed in addition to accuracy in the objective task, while there was no speed instruction for the PAS ratings as it is typical for ratings in general. In contrast, our instructions did not emphasize the speed of the response neither for objective performance nor for PAS ratings. It is possible that the speed instruction in earlier studies additionally induced responding based upon unconscious response priming in the objective task (Vorberg et al. 2003).

---

## [Decision Letter · Decision Letter 1]

12 Sep 2023

PONE-D-23-03662R1Subjective and objective measures of visual awareness convergePLOS ONE

Dear Dr. Kiefer,

Thank you for submitting your manuscript to PLOS ONE. After careful consideration, we feel that it has merit but does not fully meet PLOS ONE’s publication criteria as it currently stands. Therefore, we invite you to submit a revised version of the manuscript that addresses the points raised during the review process. Editor comments: Both reviewers have commented on the revised manuscript, noting considerable improvements. R1 advocates for publication in its current state, whereas R2 has remaining issues and insists on demanding additional evidence substantiating the equivalence of subjective (introspective) and objective measures. Although I am not an authority in your field, I find R2's critique compelling in the main proposal. While additional empirical evidence would clearly fortify your manuscript, I also see clearly that your work is already robust in its present form. Therefore, I suggest that you carefully consider R2's final remarks and determine to what extent they should be addressed, either through further evidence or through bolstering the existing arguments. I anticipate that no additional review rounds will be necessary, and I shall make a decision following the resubmission of the manuscript. Below I have some more detailed comments of my own that are aimed to help you in the final preparation of the manuscript.

We look forward to receiving your revised manuscript.

Kind regards,

Michael B. Steinborn, PhD

Section Editor

PLOS ONE

Journal Requirements:

**Additional Editor Comments:**

Firstly, I wish to clarify that my comments are intended to enhance the quality of the manuscript. While I am not the foremost expert in your specific field, my view may offer a scholarly perspective from slightly outside your domain, that is, from the view of an interested reader that stands as proxy for a broader audience. I do not expect you to adhere strictly to my remarks, however, you may find certain points to be of use of value, that exactly is my intent.

(-1-) overly lengthy discussion

Although the manuscript is generally well-written and concise, this ceases to be the case at the onset of the discussion section. While I find the discussion to be insightful and thoughtfully debated, it is somewhat lengthy, and overly focused on technical aspects, in my estimation. Consequently, my recommendation would be to revise and streamline the discussion section to make it more succinct.

(-2-) objective and subjective measures are different processes

While I am not an expert in the field, it appears to me that the question of whether objective and subjective measures represent different processes could potentially be addressed a priori through meticulous analysis of the necessary components and characteristics of the process in question. To name an example of what I mean, I suggest some specific literature that did such an analysis of the relationship between introspective and performative outcomes (doi:10.3389/fpsyg.2022.867978, chap. 4.3., 4.4, 4.5). Empirical verification in this context may be quite challenging, given that it relies on the design of the experiment and the specific factors manipulated. In the case of a temporal 2-AFC task, for instance, there are crucial factors that naturally influence the awareness threshold. At this juncture, I would expect a more comprehensive analytical exploration into the matter.

(-3-) subjective and objective measures within same trial

Your arguments to acknowledge a potential criticism—that the convergence of measures might be due to the fact that both subjective and objective measurements were collected within the same trial, could be reconsidered. I would clearly agree with the authors that pointing out that previous research, which also collected data within the same trial, did not find such convergence, in and in this way, the present research corroborates previous findings, which could be an argument that no more data collection would be needed to corroborate the present finding. In my layman perspective, however, I feel that this aspect of design may indeed by crucial here. I name an example from another field: when introspecting inattention in reaction-time series (see my aforementioned suggested review paper, around chap. 4.1 to 4.5), the self-ratings are typically given as probe trials as performance is sensitive to the frequency of "asking introspectively". Maybe it is more than to be in line with the other findings, but maybe you have some ideas of design features could be improved in future studies to - in my layman view - be better.

(-4-) slopes for easy and complex tasks

One crucial observation the authors point on is that shallower slopes (i.e., wider widths) were associated with more complex tasks (e.g., discrimination vs. detection). This result supports the notion that the transition from unawareness to awareness may differ based on the complexity of the perceptual features being evaluated. This is an absolutely interesting and in my view crucial theoretical argument that merit stronger emphasis, at least I feel it so by impulse. My immediate association is a recent work of Cao et al. (doi:10.1007/s00221-020-05861-4) who argued in the same direction supporting the author's theoretical proposal, namely that even the complexity of the rather motoric task or movement has a fundamentally altering effect on individual's introspective judgment. I suggest giving this theoretical argument more elaboration in the final revision.

(-5-) take home message

Although I am not an expert, I must express some reservations regarding the final conclusions. While there is no doubt that the findings are the result of an outstanding empirical study, the theoretical conclusions—namely, that the study challenges the traditional dichotomy between access and phenomenal consciousness—merit scrutiny. My scepticism arises from the belief that the design methodology (typically accepted within the field) may not be sufficiently critical to reveal these rather subtle differences, and this exactly is what future research should be more aware of.

Reviewers' comments:

Reviewer's Responses to Questions

**Comments to the Author**

1. If the authors have adequately addressed your comments raised in a previous round of review and you feel that this manuscript is now acceptable for publication, you may indicate that here to bypass the “Comments to the Author” section, enter your conflict of interest statement in the “Confidential to Editor” section, and submit your "Accept" recommendation.

Reviewer #1: All comments have been addressed

Reviewer #2: (No Response)

2. Is the manuscript technically sound, and do the data support the conclusions?

Reviewer #1: (No Response)

Reviewer #2: Partly

3. Has the statistical analysis been performed appropriately and rigorously? 

Reviewer #1: (No Response)

Reviewer #2: Yes

4. Have the authors made all data underlying the findings in their manuscript fully available?

Reviewer #1: (No Response)

Reviewer #2: Yes

5. Is the manuscript presented in an intelligible fashion and written in standard English?

Reviewer #1: (No Response)

Reviewer #2: Yes

6. Review Comments to the Author

Reviewer #1: (No Response)

Reviewer #2: Review of PLOS R1

Overall, I think the authors have revised this manuscript thoroughly and carefully. They have responded extensively to all of my previous comments. I think the additional explanations in the methods section help clarify the procedure. I like the newly added Fig 3, as it is very illustrative of the effects. I am still not so strongly convinced by the null effect on which the main interpretation of the results rests, as the final sample was rather small (N=20) and the power analysis was based on the assumption of a rather large effect size. In addition, the null effect is very unexpected based on the results of previous studies and thus stands quite isolated in the literature. The explanations of the authors for this divergence are plausible but purely speculative at this point. The novel analysis on the order effects does not exactly help, either, as the pattern of thresholds from the subjective ratings do not follow the objective thresholds, but given the small trial numbers I agree that this should not be strongly interpreted (and it is also questionable if the order bias in the objective thresholds would be accessible to subjective report). So I am still a bit ambiguous of this manuscript - on the one hand I find the approach and the results interesting (and of course I am not opposed to publishing null results). On the other hand, I feel that the theoretical contribution of this manuscript would be so much stronger if the null results were replicated in a second, more highly-powered, experiment; or if the authors attempted to empirically explore the reasons for the divergence of their and previous results, so publishing the work at this stage feels a bit like a missed opportunity for a much higher-impact contribution.

I spotted a few typos and other minor points:

- P. 3: …on a four-point rating scales…-> scale

- P. 16: … we did not sufficiently collected data…-> collect a sufficient number of data points

- P. 16: to empirically confirm guessing rate of 2AFC -> a guessing rate for the 2AFC task

- P. 19: for low contrast condition compared to high contrast condition -> the low, the high

- P. 20: compared to discrimination task -> to the discrimination task

- P. 24: subjective ratings threshold were -> thresholds

- P. 25: we analyzing thresholds -> analyzed

- P. 30: could also test, whether the thresholds would still converge, if the … -> remove commata

- Fig 2: shouldn’t the x-axis on the first column be labeled “trial” instead of “run #”

7. PLOS authors have the option to publish the peer review history of their article (what does this mean?). If published, this will include your full peer review and any attached files.

Reviewer #1: No

Reviewer #2: No

---

## [Author Response · Author response to Decision Letter 1]

19 Sep 2023

Responses to the Editor and to the Reviewers

We would like to thank the editor and the reviewers for their positive evaluation of our manuscript and the thoughtful and stimulating comments. We are happy to see that we successfully addressed all major concerns. Reviewer 1 was fully satisfied with our revisions, reviewer 2 had some minor points, while the editor made further suggestions for improvements. When preparing the revision, we have carefully taken all comments into account and have changed the manuscript accordingly. All changes are marked in the manuscript in yellow color.

Editor

(-1-) overly lengthy discussion

Although the manuscript is generally well-written and concise, this ceases to be the case at the onset of the discussion section. While I find the discussion to be insightful and thoughtfully debated, it is somewhat lengthy, and overly focused on technical aspects, in my estimation. Consequently, my recommendation would be to revise and streamline the discussion section to make it more succinct.

Response: Thank you for evaluating our manuscript as well-written and concise. We agree with you that the discussion is somewhat lengthy and focused on technical aspects at places. Please note that both reviewers requested to discuss these technical issues related to the nature of the 2-AFC task, fitting of psychometric functions, interpretation of the parameters of the psychometric function, etc. in detail. However, we added the discussion of these technical issues to not only satisfy our reviewers’ concerns, but also felt that a clarification of these issues, which are frequently raised, but rarely publicly addressed by researchers, would be insightful for the readership, at least for those with some interest in methodological aspects of psychophysics or psychometrics. We therefore decided to keep these technical issues, although we are aware of the broad interdisciplinary readership of PLOS One. In order to accommodate your suggestion to streamline the discussion section, we removed redundancies or reiterations at several places (text deleted between lines 716-717, and between lines 732-747). We hope that these changes will make the discussion section more concise.

(-2-) objective and subjective measures are different processes

While I am not an expert in the field, it appears to me that the question of whether objective and subjective measures represent different processes could potentially be addressed a priori through meticulous analysis of the necessary components and characteristics of the process in question. To name an example of what I mean, I suggest some specific literature that did such an analysis of the relationship between introspective and performative outcomes (doi:10.3389/fpsyg.2022.867978, chap. 4.3., 4.4, 4.5). Empirical verification in this context may be quite challenging, given that it relies on the design of the experiment and the specific factors manipulated. In the case of a temporal 2-AFC task, for instance, there are crucial factors that naturally influence the awareness threshold. At this juncture, I would expect a more comprehensive analytical exploration into the matter.

Response: Thank you for suggesting a more comprehensive analytical exploration of commonalities and differences of objective and subjective measures of awareness and for drawing our attention to the interesting review article on cognitive restauration by Schumann and colleagues, which is now cited in our manuscript. In the previous version of the manuscript, we have already provided detailed analyses regarding the processes underlying objective and subjective measures, possible commonalities and differences and associated measurement problems. In fact, the analytical cognitive explorations provided in the manuscript lead to the three possible scenarios regarding the relation between objective and subjective measures of awareness described at the end of the introduction section (independence of objective vs. subjective measures, relation, but constant lag, zero lag). Perhaps, these analytical cognitive explorations were not particular salient in the previous version of our manuscript, because they were provided at various places in the introduction section. As a first measure, we moved the definitions of access vs. phenomenal consciousness from the end to the manuscript to the beginning of the introduction section (p. 4, lines 68-73). Furthermore, in order to make the connection between the three scenarios of a relation between objective and subjective measures of awareness and the analytical cognitive explorations in the preceding paragraphs of the manuscript more salient, we shortly repeat the key findings or theoretical arguments related to a given scenario at the end of the introduction section (p. 11, lines 247-263): “With regard to the relation between objective performance and subjective PAS ratings measures, depending on the precise theoretical stance, three alternative hypothetical scenarios are possible: Firstly, thresholds derived from objective and subjective measures of awareness could be unrelated across tasks and contrasts, because these measures might capture qualitatively different forms of information or representations [31,32] linked to access vs. phenomenal consciousness [1,43]. Objective and subjective measures may also partially depend on different types processes [for a discussion, also see 44]: For instance, only subjective, but not objective measures might depend on second-order meta-cognitive processes, which evaluate the representation [33]. Secondly, objective and subjective thresholds could exhibit a comparable pattern as a function of task and contrast, but subjective thresholds would be temporally delayed by a constant lag. This lag may arise because above-threshold objective performance might be partially based on fast unconscious processing, while above-threshold subjective ratings may require longer lasting visual consolidation giving rise to a specific phenomenal experience [10,18,28]. Thirdly, PAS ratings and accuracy of task performance could comparably reflect the phenomenal content of visual awareness. As a consequence, thresholds should be similar for subjective and objective measures, exhibiting a zero-lag.”

(-3-) subjective and objective measures within same trial

Your arguments to acknowledge a potential criticism—that the convergence of measures might be due to the fact that both subjective and objective measurements were collected within the same trial, could be reconsidered. I would clearly agree with the authors that pointing out that previous research, which also collected data within the same trial, did not find such convergence, in and in this way, the present research corroborates previous findings, which could be an argument that no more data collection would be needed to corroborate the present finding. In my layman perspective, however, I feel that this aspect of design may indeed by crucial here. I name an example from another field: when introspecting inattention in reaction-time series (see my aforementioned suggested review paper, around chap. 4.1 to 4.5), the self-ratings are typically given as probe trials as performance is sensitive to the frequency of "asking introspectively". Maybe it is more than to be in line with the other findings, but maybe you have some ideas of design features could be improved in future studies to - in my layman view - be better.

Response: Thank you again for drawing our attention to the review paper on cognitive restauration by Schumann and colleagues, which discusses possible interactions between objective performance and introspection. We entirely agree that such interactions can in principle take place. In fact, we have already investigated such interactions in a previous study (Kiefer, M., Harpaintner, M., Rohr, M., & Wentura, D. (2023). Assessing subjective prime awareness on a trial-by-trial basis interferes with masked semantic priming effects. Journal of Experimental Psychology: Learning, Memory, and Cognition, 49(2), 269-283. doi: 10.1037/xlm0001228). However, with regard to the present study, we do not see how such interactions could induce a common threshold of awareness. Nevertheless, we added a short discussion of within-trial interactions including a citation of the Schumann et al. paper (p. 25, lines 574-579). Please note that in the previous revision round in response to reviewer 2 we already provided suggestions for design changes, in order to systematically assess such interactions. In the new discussion paragraph, we foreshadow these suggested design changes, which include collection of subjective and objective measures in different blocks of trials: ” Although introduction of subjective ratings can in principle influence performance in the same [52] or in the next trial [44], we do not see how interactions between subjective and objective awareness tasks could induce a convergence of thresholds. Nevertheless, to specifically examine cross-task interactions, future studies could collect subjective and objective awareness measures in different trials (for a further discussion, see below).”

(-4-) slopes for easy and complex tasks

One crucial observation the authors point on is that shallower slopes (i.e., wider widths) were associated with more complex tasks (e.g., discrimination vs. detection). This result supports the notion that the transition from unawareness to awareness may differ based on the complexity of the perceptual features being evaluated. This is an absolutely interesting and in my view crucial theoretical argument that merit stronger emphasis, at least I feel it so by impulse. My immediate association is a recent work of Cao et al. (doi:10.1007/s00221-020-05861-4) who argued in the same direction supporting the author's theoretical proposal, namely that even the complexity of the rather motoric task or movement has a fundamentally altering effect on individual's introspective judgment. I suggest giving this theoretical argument more elaboration in the final revision.

Response: Thank you for recommending the paper by Cao and colleagues on action binding and the time course of the perceived action depending on action complexity. It is indeed intriguing that a similar more gradual transition from unawareness to awareness for complex vs. simple features has been suggested for somatosensory features associated with actions. As this editor mentions in his point 1 that the discussion section is already quite long, we added just two short sentences to highlight this aspect (pp. 26-27, lines 621-625): “Interestingly, a similar more gradual transition from unawareness to awareness for complex vs. simple features has been suggested for somatosensory features associated with actions [55]. Possibly, a more gradual emergence of awareness for complex features might be a functional principle also in other domains than vision.”

(-5-) take home message

Although I am not an expert, I must express some reservations regarding the final conclusions. While there is no doubt that the findings are the result of an outstanding empirical study, the theoretical conclusions—namely, that the study challenges the traditional dichotomy between access and phenomenal consciousness—merit scrutiny. My scepticism arises from the belief that the design methodology (typically accepted within the field) may not be sufficiently critical to reveal these rather subtle differences, and this exactly is what future research should be more aware of.

Response: Please note that we have never argued that our study challenges the traditional dichotomy between access and phenomenal consciousness. As these types of consciousness essentially differ with regard to their perspective (third vs. first person perspective), this dichotomy cannot be resolved, even if objective and subjective measures of awareness converge. We have only argued that access and phenomenal consciousness possibly rely on the same representations. As this statement might be misleading and could evoke the impression that we question the dichotomy of access and phenomenal consciousness, we have deleted in the discussion section all statements (p. 31, lines 732-747), which could evoke connotations in this direction. This has the further advantage that the final conclusion section is much shorter now in line with the editor’s point 1. It now reads (p. 31, lines 741-747): In conclusion, using a temporal 2-AFC task we found a comparable pattern of thresholds across tasks and contrasts for objective and subjective measurements of awareness. This finding suggests that objective performance measures based on accuracy and subjective ratings of the visual experience can provide similar information on the feature-content of a percept [see also, 51]. The observed similarity of thresholds validates both psychophysical and subjective approaches to awareness as converging and thus most likely veridical measures of awareness.

Reviewer 1

All comments have been addressed

Response: We are glad to see that we have addressed this reviewer’s points satisfactorily.

Reviewer 2

Overall, I think the authors have revised this manuscript thoroughly and carefully. They have responded extensively to all of my previous comments. I think the additional explanations in the methods section help clarify the procedure. I like the newly added Fig 3, as it is very illustrative of the effects. I am still not so strongly convinced by the null effect on which the main interpretation of the results rests, as the final sample was rather small (N=20) and the power analysis was based on the assumption of a rather large effect size. In addition, the null effect is very unexpected based on the results of previous studies and thus stands quite isolated in the literature. The explanations of the authors for this divergence are plausible but purely speculative at this point. The novel analysis on the order effects does not exactly help, either, as the pattern of thresholds from the subjective ratings do not follow the objective thresholds, but given the small trial numbers I agree that this should not be strongly interpreted (and it is also questionable if the order bias in the objective thresholds would be accessible to subjective report). So I am still a bit ambiguous of this manuscript - on the one hand I find the approach and the results interesting (and of course I am not opposed to publishing null results). On the other hand, I feel that the theoretical contribution of this manuscript would be so much stronger if the null results were replicated in a second, more highly-powered, experiment; or if the authors attempted to empirically explore the reasons for the divergence of their and previous results, so publishing the work at this stage feels a bit like a missed opportunity for a much higher-impact contribution.

Response: We thank this reviewer for acknowledging our effort in addressing their concerns and in improving our manuscript. We feel honored that this reviewer acknowledges the theoretical importance and empirical soundness of our study. It is true that replication of the present findings and exploration of suggested reasons for divergent findings are important. We would like to mention at this place that somewhat related to the present work, a recent study using constant mask stimulus SOAs, just published a few weeks ago, found comparable objective and subjective d’ sensitivity measures. Hence, our observation of convergence of objective and subjective measures is not so isolated any more. We now mention this study in the revised manuscript on p. 24, lines 545-549. Given that so many new issues could and should be addressed in future studies, we felt that adding one new experiment to this manuscript exploring one question would still appear somewhat unsatisfactory because other 3 or 4 questions would have still not been addressed. We certainly share the interest with this reviewer to get these new questions answered, but think it is more appropriate to do this in a future publication, given that the present work is already fairly complex

I spotted a few typos and other minor points:

- P. 3: …on a four-point rating scales…-> scale

- P. 16: … we did not sufficiently collected data…-> collect a sufficient number of data points

- P. 16: to empirically confirm guessing rate of 2AFC -> a guessing rate for the 2AFC task

- P. 19: for low contrast condition compared to high contrast condition -> the low, the high

- P. 20: compared to discrimination task -> to the discrimination task

- P. 24: subjective ratings threshold were -> thresholds

- P. 25: we analyzing thresholds -> analyzed

- P. 30: could also test, whether the thresholds would still converge, if the … -> remove commata

- Fig 2: shouldn’t the x-axis on the first column be labeled “trial” instead of “run #”

Response: Thank you for highlighting these typos and minor points. The manuscript has been changed as suggested, including the label on the abscissa in the first column of Fig 2 (“trial”).

---

## [Editor Report · Decision Letter 2]

20 Sep 2023

Subjective and objective measures of visual awareness converge

PONE-D-23-03662R2

Dear Dr. Kiefer,

We’re pleased to inform you that your manuscript has been judged scientifically suitable for publication and will be formally accepted for publication once it meets all outstanding technical requirements.

Kind regards,

Michael B. Steinborn, PhD

Section Editor

PLOS ONE
---

## [Editor Report · Acceptance letter]

25 Sep 2023

PONE-D-23-03662R2 

Subjective and objective measures of visual awareness converge 

Dear Dr. Kiefer:

I'm pleased to inform you that your manuscript has been deemed suitable for publication in PLOS ONE. Congratulations! Your manuscript is now with our production department. 

Kind regards, 

on behalf of

Dr. Michael B. Steinborn 

Section Editor

PLOS ONE